# miR-142: A Master Regulator in Hematological Malignancies and Therapeutic Opportunities

**DOI:** 10.3390/cells13010084

**Published:** 2023-12-30

**Authors:** Wilson Huang, Doru Paul, George A. Calin, Recep Bayraktar

**Affiliations:** 1Department of Translational Molecular Pathology, The University of Texas MD Anderson Cancer Center, Houston, TX 77030, USA; whuang58@jhu.edu (W.H.); gcalin@mdanderson.org (G.A.C.); 2Division of Hematology and Medical Oncology, Department of Medicine, Weill Cornell Medicine, New York, NY 10065, USA; dop9054@med.cornell.edu; 3Center for RNA Interference and Non-Coding RNAs, The University of Texas MD Anderson Cancer Center, Houston, TX 77030, USA; 4Department of Leukemia, Division of Cancer Medicine, The University of Texas MD Anderson Cancer Center, Houston, TX 77030, USA

**Keywords:** non-coding RNA, microRNA, miR-142, physiology, hematopoiesis, exosome, mutation, cancer, lymphoma, leukemia

## Abstract

MicroRNAs (miRNAs) are a type of non-coding RNA whose dysregulation is frequently associated with the onset and progression of human cancers. miR-142, an ultra-conserved miRNA with both active -3p and -5p mature strands and wide-ranging physiological targets, has been the subject of countless studies over the years. Due to its preferential expression in hematopoietic cells, miR-142 has been found to be associated with numerous types of lymphomas and leukemias. This review elucidates the multifaceted role of miR-142 in human physiology, its influence on hematopoiesis and hematopoietic cells, and its intriguing involvement in exosome-mediated miR-142 transport. Moreover, we offer a comprehensive exploration of the genetic and molecular landscape of the miR-142 genomic locus, highlighting its mutations and dysregulation within hematological malignancies. Finally, we discuss potential avenues for harnessing the therapeutic potential of miR-142 in the context of hematological malignancies.

## 1. Introduction

Non-coding RNAs (ncRNAs) make up the majority of the human transcriptome and play diverse roles in transcriptional, translational, and epigenetic regulation [1,2,3]. ncRNAs are generally classified into small ncRNAs, with lengths of less than 200 ribonucleotides, and long ncRNAs, with lengths of more than 200 ribonucleotides (some exceeding 90,000) [4,5,6]. Small ncRNAs include microRNAs (miRNAs), small interfering RNAs (siRNAs), small nucleolar RNAs (snoRNAs), small nuclear RNAs (snRNAs), transfer RNA (tRNA)-derived fragments, and Piwi protein-interacting RNAs (piRNAs) [4,5]. miRNAs, typically 20 to 25 ribonucleotides in length, are the most extensively studied type of small ncRNAs; they have been found to regulate over 60% of human genes [7,8].

The biogenesis of miRNAs first involves transcription via RNA polymerase II/III to form primary miRNAs (pri-miRNAs) with interspersed mismatched nucleotides and a hairpin structure [9,10,11]. Pri-miRNAs are then cleaved at the base of the hairpin structure by the RNA-binding protein DGCR8 and the enzyme Drosha to generate shortened precursor miRNAs (pre-miRNAs). The pre-miRNAs are released outside the nucleus through the facilitation of an exportin-5/Ran-GTP complex [12,13]. In the cytoplasm, pre-miRNAs are cleaved by the RNA-binding protein TRBP and the enzyme Dicer to eliminate the hairpin loop, leaving a short, mature miRNA duplex [9,12]. Directed by what the cell type or environment necessitates, the 5p strand, 3p strand, or both strands of the mature miRNA duplex may be incorporated into an Argonaute (AGO) family protein in the RNA-induced silencing complex (RISC). Unused strands are subsequently degraded by cellular mechanisms [11,14]. In most cases, miRNAs facilitate the binding of RISC to the 3′ untranslated region (3′UTR) of mRNAs, which allows for mRNA degradation through perfect complementary binding or translational inhibition through partial complementary base pairing [15,16].

In the last two decades, the implications of miRNA dysregulation in human cancer have garnered significant attention, presenting opportunities to harness miRNA-based cancer diagnostics, prognostics, and therapeutics [17,18,19,20]. In particular, miR-142, a miRNA expressed in various tissues, but preferentially among hematopoietic cells, has been systematically studied due to its role in many solid tumors, such as breast, renal, and colorectal cancers, but more notably in hematological malignancies, such as lymphoma and leukemia [21,22,23,24,25,26,27,28,29]. Highly conserved in vertebrates, miR-142 targets critical genes, including *MRFAP1*, *OSBP*, *NFE2L2*, *B2M*, *HERPUD1*, *ARNTL*, *CCNT2*, *LGR5*, *FLT1*, *PROM1*, *ABCG2*, *TAB2*, *MCL1*, *PAPD5*, and *UBE2A* genes, controlling cellular functions such as apoptosis, proliferation, immune response, detoxification, viral interaction, and tumorigenesis [30,31,32,33,34,35,36,37,38].

In this review, we briefly discuss the biogenesis of miR-142, its regulators, and the functions it carries out under such regulation. We narrow our focus to the role of miR-142 in hematopoietic cells and the exosome-mediated transport of miR-142 among these cells. We then discuss the relationship between miR-142 and hematological cancers, including multiple types of lymphomas and leukemias, through the lens of mutations, altered cellular regulation pathways, and aberrant expression levels. We conclude by briefly presenting possible therapeutic avenues for targeting miR-142, as exemplified by current miRNA drugs and DNA-based biotechnology. We strongly believe that enhancing the scientific community’s understanding of the complex and often intertwined roles of miR-142 in hematopoietic physiology and hematological malignancies will aid efforts to launch current therapeutic platforms for miR-142-based treatment while reducing unanticipated side effects.

## 2. Biogenesis, Regulation, and Functions of miR-142

### 2.1. Biogenesis of miR-142

The components of miR-142 are illustrated in the context of the pre-miR-142 sequence and structure in Figure 1, and the biogenesis of miR-142 is summarized in Figure 2. The miR-142 sequence is conserved in the *MIR142HG* gene, which is commonly transcribed as a long ncRNA and located at chromosome 17q22 [21,39]. A particularly unique feature of miR-142 that separates it from other miRNAs is the fact that both mature miR-142-3p and -5p strands are highly conserved among vertebrates, such as rats, mice, and humans, and both strands play an active role in gene regulation [28,40].

Ultraconserved regions are completely conserved (100%) across orthologous regions of the human, rat, and mouse genomes [41]. A substantial proportion of these ultraconserved regions are located within ncRNA regions. Notably, although ultraconserved miR-142-3p and -5p exhibit shared targets, in many cases, these two strands independently modulate distinct targets, either by inhibiting or derepressing them [40]. Seed sequences are the first 7/8 nucleotides within the miR-142-3p and -5p strands that act as binding sites for the 3′UTR of mRNA. As will be explored, the control of miR-142 expression frequently intertwines with the requirement for mature miR-142-3p and -5p strands to directly affect specific targets while also triggering essential downstream consequences.

### 2.2. Regulation and Function of miR-142

Regulators of miR-142 expression are dependent on the organ under investigation. In the brain, CLOCK/BMAL1 proteins serve as positive regulators of miR-142 transcription, elevating levels of miR-142-3p. miR-142-3p binds to the 3′UTR of *Bmal1* mRNA, reducing BMAL1 protein levels. This mechanism forms an essential negative feedback loop in the maintenance of the circadian rhythm [34]. In the heart, however, MAPK and p300 act as negative regulators of miR-142 transcription, resulting in the downregulation of both miR-142-3p and -5p. A mutual inhibitory feedback loop between miR-142-5p and *p300* is formed through the binding of miR-142-5p to the 3′UTR of *p300* mRNA. The downregulation of both mature miR-142 strands induces the expression of genes such as α-actinin and the activation of widespread cytokine signaling for myocyte growth, survival, and functionality during cardiac hypertrophy [42]. miR-142-3p is also negatively regulated by a *TUG1* long ncRNA post-transcriptionally during myocardial injury. The depletion of miR-142-3p releases the suppression of *HMGB1* and *Rac1* mRNA at the 3′UTR, stimulating autophagy and apoptosis [43].

### 2.3. Hematopoietic Functions of miR-142

Although miR-142 plays various roles in the brain, heart, bone, eye, kidney, liver, lung, pancreas, and skin, it is more notably expressed in hematopoietic cells [34,42,44,45,46,47,48,49,50]. Sun et al. found that the PU.1, Runx1, and C/EBP-beta (C/EBPβ) proteins act as transcription factors that can bind to the miR-142 promoter to induce miR-142 transcription [51]. Specifically, PU.1, independently or with the help of Runx1, CBF-beta (CBFβ), and C/EBPβ, maintains a dominant force in miR-142 regulation [51]. Along with other mechanisms, miR-142 expression levels are tightly controlled to facilitate the proper differentiation, growth, and function of all cells originating from the hematopoietic stem cell, including key myeloid and lymphoid cells in the body. The impacts of miR-142 dysregulation are therefore evident in abnormalities in the blood and defects in the immune system. These impacts are summarized in Figure 3 and described below.

#### 2.3.1. Hematopoiesis

Hematopoiesis is the process by which all blood cells, including erythrocytes, leukocytes, and thrombocytes, are developed through hematopoietic stem cell differentiation [52]. The miR-223–C/EBPβ–LMO2–miR-142 signaling pathway serves an essential regulatory function in hematopoiesis [53]. This is an instance in which a miRNA (miR-223) indirectly regulates another (miR-142) through the actions of C/EBPβ and LMO2 nuclear proteins [53]. C/EBPβ is part of a family of transcription factor proteins with a shared structure consisting of a C-terminal leucine zipper, DNA-binding site, and N-terminal transactivation region [54]. When the leucine zipper is dimerized, C/EBPβ initiates DNA binding and generates a cascade effect [54]. C/EBPβ transcriptionally and miR-223 post-transcriptionally downregulate levels of LMO2-L and LMO2-S [53]. Functioning as isoforms of LMO2 and transcription factor proteins, LMO2-L and LMO2-S negatively control miR-142 production [55]. As a hematopoietic-specific miRNA, miR-142 ultimately functions to restrict hematopoietic cell proliferation [53,56]. By acting through miR-142, C/EBPβ and LMO2 critically regulate the tissue- and stage-specific gene expression of myeloid and lymphoid cellular differentiation and function and determine the B- and T-cell lineage commitment and development [57,58,59,60]. The importance of LMO2 cannot be overlooked. In its role as an oncogene, an elevated level has caused T-cell growth arrest and T-acute lymphoblastic leukemia (T-ALL)-like syndrome in patients with X-linked severe combined immunodeficiency (SCID-X1) who received retroviral vectors for gene therapy [61,62]. Hence, pathways exist in hematopoietic cells to regulate miR-142 for appropriate proliferation and differentiation of hematopoietic cells.

#### 2.3.2. B Cells

B cells are lymphocytes that play a crucial role in adaptive immunity by secreting diverse antibodies. miR-142 is not only a crucial regulator of hematopoiesis but is also involved in B-cell lymphopoiesis [53,63]. The importance of miRNAs to B-cell lineage survival, differentiation, tolerance, and antibody diversity and for germinal center B-cell formation was recognized early on from *Dicer*-conditional-deletion models [64,65,66]. In *mb1*^(*Cre*/+)^*Dicer*^(*fl*/*fl*)^ mice with *Dicer* gene deletion, almost all B-cell development is stuck at the early pro-B-cell stage due to BIM-mediated apoptosis, and antibody diversity broadens toward the Igκ variable domains [64]. In *CD19-Cre*^(*ki*/+)^*Dicer*^(*fl*/*fl*)^ mice modeling late B-cell differentiation, the marginal-zone B cells in lymphoid follicles expand, whereas the generation of follicular-zone B cells is hampered [65]. These Dicer-deficient B cells have a biased B-cell antigen receptor (BCR) repertoire and produce autoreactive antibodies with autoimmune attributes to break self-tolerance [65]. In an *Aicda*^(*Cre*/+)^*Dicer*^(*fl*/*fl*)^ mouse model focusing solely on activated B cells, the B cells in the germinal center fail to form properly due to deficits in cell proliferation and viability, resulting in impaired generation of memory B cells and plasma cells required for high-affinity class-switched antibody formation [66].

The additional impact of miR-142 on B-cell lymphopoiesis and homeostasis for the generation of adaptive immunity can be directly seen in miR-142-deficient mice [63]. The phenotype consequences of miR-142 deficiency resemble *Dicer* conditional deletions. B cells in the marginal zone expand, develop abnormally, and fail to generate plasma cells to secrete immunoglobulin, while the B1 B-cell and T-cell populations in the peripheral peritoneal cavity shrink, resulting in severe combined immunodeficiency [63]. In essence, mature humoral B-cell and cellular T-cell functions vital for adaptive immunity are obstructed by an inability to mount an antibody response to T-cell-dependent antigens [63]. The miR-142-deficient mouse study also demonstrates and supports the findings that miR-142-3p, not miR-142-5p, is the central miRNA regulating immune cells, as evidenced by the scope of miR-142-3p targeted mRNA derepression [63,67,68]. One such target is the B-cell-activating factor receptor (BAFF-R), an important signaling molecule on the surface of B cells that, together with BCR, regulates B-cell survival and homeostasis [63,69]. These findings highlight the importance of miR-142 in B-cell lymphopoiesis by maintaining B-cell populations and enabling plasma cells to secrete immunoglobulins for immunity.

#### 2.3.3. T Cells

T cells are lymphocytes that make up another key component of adaptive immunity through their recognition and cytotoxicity against specific antigens. miRNA control and regulation are also critical to T-cell development, homeostasis, and function. *Dicer* deletion is again useful for the interrogation of miRNA functions. In the T-cell-specific *Dicer*-conditional-deletion model, naïve CD4+ helper T cell and CD8+ cytotoxic T-cell development are defective in the thymus, resulting in reduced peripheral T-cell numbers in the spleen, lymph nodes, and blood [70]. These CD4+ helper T cells differentiate abnormally and preferentially produce the cytokine IFN-gamma (IFNγ); they also grow poorly and undergo increased apoptosis upon stimulation [70]. In miR-142-deficient mice, CD4+ and CD8+ T cells also drastically decrease in the spleen, but the ratio between CD4+ and CD8+ T cells is not affected [63]. These peripheral T cells are impaired in typical T-cell receptor-driven proliferation, such as CD3 and CD28 stimulation [71]. Interestingly, T-cell development in the central thymus is normal [63]. These features highlight that miR-142 is indispensable for mature T cell survival and distribution to peripheral compartments and their function within the lymphoid organs.

One of the important CD4+ helper T-cell functions is that of regulatory T (Treg) cells, which are regulators of immune responses pertaining to tolerance, autoimmunity, and anti-tumor immunity [72]. Upon transforming growth factor beta (TGFβ) cytokine stimulation and induction of transcription factor Foxp3, peripheral CD4+CD25− naïve T cells convert to CD4+CD25+ Treg cells [73]. These Treg cells possess the ability to suppress immunological responses to themselves through TGFβ, IL-10, and IL-35 cytokine secretion [72]. Notably, in miR-142-deficient mice with global miR-142 ablation, the Treg cell population decreases drastically both in the peripheral lymphoid organs and in the central thymus [74]. In *Foxp3*^(*Cre*)^*miR-142*^(*fl*/*fl*)^ mice, which exhibit miR-142 ablation in local Treg cells, a lethal and systemic autoimmune disease arises. This autoimmunity is characterized by lymphadenopathy, splenomegaly, thymic shrinkage, skin rash, and overwhelming infiltration of CD4+ memory T cells and CD8+ effector T cells into the peripheral tissues [74]. Mechanistically, ablation of miR-142-3p in Treg cells results in a derepression of IFNγ-associated genes, such as *Ifngr2*, *Gbp3*, *Stat1*, *Irf1*, and *Hif1a*, resulting in excessive INFγ production by Treg, CD4+, and CD8+ T cells [74]. IFNγ is a key cytokine in anti-tumor immunity [75]. This excessive IFNγ increases the “fragility” of Treg cells, causing autoimmunity and, conversely, enhancing anti-tumor immunity [76]. These findings establish that miR-142 serves a critical regulatory role for Treg-cell development, function, and homeostasis.

#### 2.3.4. Natural Killer Cells

Natural killer (NK) cells belong to the innate lymphocyte (ILC) family, together with type I ILCs (ILC1s), and derive from the same lymphoid lineage as B and T cells [77]. While B and T cells constitute the body’s adaptive immunity, NK cells are involved in innate immunity against external pathogens and endogenous tumors through the production of perforin, granzymes, and the cytokines IFNγ and tumor necrosis factor alpha (TNFα) [78,79]. Under normal development, NK cell precursors and NK cells rely on cell surface IL-15 receptor engagement with the cytokine IL-15 to maintain development, function, and homeostasis [80]. miR-142-deficient mice display a reduction in NK cells in peripheral spleen, blood, and lymph nodes due to diminished IL-15 receptor signaling and an increase in ILC1-like cells reflective of the simultaneous rise in TGFβ involvement [81,82]. The local presence of TGFβ and increased TGFβ signaling within a tissue, including the tumor microenvironment, can transform NK cells into ILC1-like cells, a mechanism by which tumors can avoid innate immune surveillance [83]. In the bone marrow, NK cell precursors and NK cell maturation are not affected by miR-142 deficiency, but ILC1-like cells expand [81]. The common lymphoid progenitor (CLP), which is the lineage source for B cells, T cells, NK cells, and ILCs, is not altered by miR-142 deficiency [81]. In miR-142-deficient mice, both NK cells and ILC1-like cells exhibit lower IFNγ production upon stimulation and fail to generate TNF-related apoptosis-inducing ligand (TRAIL)-expressing target cell killing [81]. These findings demonstrate that miR-142 is essential for peripheral NK cell homeostasis and innate effector functions under the influence of the local milieu [82]. For example, in nasal natural killer/T-cell lymphoma (NKTCL), which is commonly linked to Epstein–Barr virus (EBV), miR-142-3p is downregulated and IFNγ production is inhibited, thereby promoting lymphoma transformation with increased oncogenic BCL6 and IL1A cytokine expression [84,85]. As evidenced by the drop in the NK cell population and the defective NK killing mechanisms that result from miR-142 deficiency, miR-142 is indispensable for proper NK cell lymphopoiesis and function.

#### 2.3.5. Myeloid Cells

Unlike lymphoid cells, myeloid cells (including granulocytes, monocytes, macrophages, and dendritic cells) play an active role in innate immunity [86]. miR-142 is the principal gatekeeper in determining the lineage pathway for lymphoid and myeloid differentiation [53,59]. During normal myelopoiesis, common myeloid progenitor cells differentiate into granulocytes, such as basophils, neutrophils, and eosinophils; monocytes, such as macrophages; and dendritic cells—which together constitute the innate immune system [87]. In miR-142-deficient mice, this process is compromised, resulting in significant amplification of the myeloid cell population and abnormally high infiltration of myeloid cells into the spleen, disrupting normal germinal center formation and function [63]. This is due to the shifting of the multipotent hematopoietic progenitors to the myeloid lineage as the lymphoid lineage is repressed [27]. As we discuss in Section 3.2, miR-142-3p expression increases upon all-*trans* retinoic acid (ATRA)-induced granulocytic differentiation and phorbol myristate acetate (PMA)-induced monocytic differentiation in leukemia cell lines [36]. Without ATRA and PMA, miR-142-3p mimics alone have also been shown to drive granulocytic and monocytic differentiation, whereas anti-miR-142-3p inhibitors have the opposite effects [53]. Similarly, such miRNA-142-3p overexpression and knockdown effects can also be reproduced using CD34+ hematopoietic stem/progenitor cells cultured from the umbilical cord blood and bone marrow samples of healthy individuals [36]. These findings show that miR-142 is important in the regulation of normal myeloid differentiation.

### 2.4. Exosome-Mediated Transport of miR-142

Exosomes are 30 to 100 nm extracellular vesicles that are released from all cell types via multivesicular endosome fusion with the plasma membrane and function in intercellular communication and signaling [88,89]. Exosomes were first discovered during the differentiation of maturing reticulocytes and later also found to be actively secreted by the lymphoid and myeloid cells of the hematopoietic system, including B cells, T cells, macrophages, mast cells, and dendritic cells [90,91,92,93,94,95,96]. Exosomes are characterized by distinct proteins engaged in membrane transport and fusion (flotillin, annexins, and Rab GTPase), multivesicular body biogenesis (Alix and TSG101), tetraspanins (CD9, CD63, CD81, and CD82), and heat-shock proteins (hsc70 and 90) [97]. So far, more than 4500 proteins have been identified in the exosome proteome via mass spectrometry, reflecting the enriched cellular proteins from which exosomes are derived [98]. The exosome membrane consists of cholesterol, ceramide, sphingolipids, ceramide, and phospholipids with long and saturated fatty-acyl chains that participate in the lipid raft functions associated with endocytosis, exocytosis, and cellular signaling [99]. Besides proteins and lipids, exosomes also carry extensive molecular cargo, including DNA, mRNA, miRNA, rRNA, tRNA, piRNA, small Cajal body-specific RNA (scaRNA), and snoRNA [93,100,101].

Intracellular trafficking for exosome biogenesis follows two pathways: the endosomal sorting complex required for transport (ESCRT)-dependent pathway or the synthesis of ceramide (ESCRT-independent) pathway. It is known that miRNAs utilize the ESCRT-independent pathway; they are loaded into endosomes, which are sorted and secreted as extracellular exosomes [102]. Experimentally blocking the ESCRT-independent pathway via inhibition of the lipid-generating enzyme nSMase2, which is responsible for ceramide synthesis, has a minimal effect on miRNA secretion [89]. Some miRNAs are sorted into the intraluminal vesicles of multivesicular bodies by a completely different mechanism. In primary T cells, sumoylated nuclear ribonucleoprotein A2B1 (hnRNPA2B1) binds to sequence motifs present in the miRNAs and regulates miRNA sorting. Sumoylation, as a post-translational modification, controls hnRNPAB1 and miRNA binding [103]. On the other hand, free circulating miRNAs, including miR-142-3p, have been reported to complex with Argonaute2 (Ago2) in the plasma for stability and protection from RNases [104]. This not only serves as an exception to the consensus model that miRNAs are transported in membrane-bound exosomes in the extracellular environment but also suggests that cells could also release a functional miRNA-inducing silencing complex into the circulation in addition to exosomes. A stoichiometric analysis of the plasma miRNA content of exosomes also supported this alternate transport route; quantitatively, far less than one molecule of a given miRNA was present per exosome, even for the most abundant miRNAs in the exosome preparation [105].

Despite the complexity of miRNA exosome sorting and transport, the functional effects of miR-142 exosomes can be seen in two representative cell types of the hematopoietic system. Acting as lymphoid cells in the adaptive immune response, T-cell-derived miR-142 exosomes play an important role in normal physiology and disease. In Type 1 diabetes, exosome-mediated transfer of miR-142-3p and miR-142-5p from T lymphocytes to pancreatic beta cells results in the expression of the inflammatory chemokines Ccl2, Ccl7, and Cxcl10; apoptosis; and death of beta cells [106]. The sequela is an autoimmune attack and the killing of pancreatic islet cells by the immune cells. In experimental autoimmune myocarditis as a counterpart of human myocarditis, internalization of abnormally elevated circulating miR-142 exosomes by CD4+ T cells causes glycolytic metabolic reprogramming and subsequent activation of the immune cells via miRNA-mediated targeting and suppression of the MBD2 and SOCS1 proteins [103]. CD4+ T cell immunometabolic dysfunction is associated with experimental autoimmune myocarditis progression [107]. In post-myocardial infarction ischemic remodeling, CD4+ T cells infiltrate into the heart and facilitate sustained inflammation. CD4+ T cell-derived exosomal miR-142-3p activates myofibroblasts through the targeting of adenomatous polyposis coli (APC) and inhibition of the WNT signaling cascade, leading to cardiac fibrosis and malfunction [108]. In Sjogren syndrome, activated T-cell-derived miR-142-3p exosomes impair the epithelial functions of the salivary and lacrimal exocrine glands by downregulating the expression of SERCA2B and RyR2 proteins, which decrease Ca2+ signaling and cAMP production [109]. In contrast, serving as myeloid cells in the innate immune response, macrophage-derived miR-142 exosomes are essential to lung physiologic function. In idiopathic pulmonary fibrosis, upregulation of microphage-derived exosome miR-142-3p inhibits the expression of TGFβ receptor (TGFβR1) and profibrotic genes in the airway epithelial cells and lung fibroblasts, thereby reducing the interstitial lung fibrosis critical to the progression of the disease [110].

These findings highlight the effect of exosomes in mediating the transport of miR-142 from hematopoietic cells, such as T cells and myeloid cells, to different organs of the body to maintain physiology. Further studies will be necessary to evaluate the functional effects of miR-142 exosomes in hematological malignancies.

## 3. The Role of miR-142 in Hematological Malignancies

The B-, T-, NK-, and myeloid-cell defects reflected in different levels of miR-142 knockout mouse models point to the essential nature of miR-142 regulation in hematopoietic cells. Hence, it is unsurprising that mutations and the aberrant expression of miR-142 are commonly involved in the development and progression of hematological malignancies. Mechanistically, much of the cellular effect of miR-142 dysregulation has been deciphered, with upregulation or downregulation of direct targets and downstream biomolecules being identified in the pathophysiology of lymphomas and leukemias. Besides presenting the possibility of utilizing miR-142 as a diagnostic biomarker, atypical expression levels of miR-142 have also been evaluated for feasibility in predicting prognosis. We here describe the current understanding of miR-142 in specific hematological malignancies.

### 3.1. Lymphomas

#### 3.1.1. Diffuse Large B-Cell Lymphoma

Diffuse large B-cell lymphoma (DLBCL) is the most common subtype of non-Hodgkin lymphoma worldwide [111,112]. Occurring either de novo or through the transformation of indolent lymphoma, DLBCL is one of the most aggressive forms of B-cell lymphoma and remains incurable for around 40% of patients [113]. A pan-cancer analysis conducted by Urbanek-Trzeciak et al. using publicly available data from The Cancer Genome Atlas (TCGA) showed that miR-142 in DLBCL had the highest frequency of mutations among overmutated miRNAs and cancers, at a rate of almost 22% [114]. In a cohort of 37 patients, three point mutations occurred in the same location in the miR-142-3p seed sequence, or “seed” [114]. Another mutation occurred in the miR-142-3p seed, two mutations occurred outside the seed but within the mature miR-142-3p strand, two mutations occurred outside the seed but within the mature miR-142-5p strand, and two mutations occurred outside both mature forms of miR-142 but within the pre-miR-142 [114]. At a comparable rate, Hezaveh et al. found three cases of miR-142 mutations in a cohort of 19 DLBCL patients [115]. In each case, a new point mutation was found in either the miR-142-3p seed, outside the seed but within the mature miR-142-3p strand, or outside the mature miR-142-3p strand but within the pre-miR-142 sequence [115]. These findings are summarized in Table 1.

Consistent with these findings, DLBCL patients exhibited a high frequency of miR-142 mutations, nearing 20% in a cohort of 56 cases from a study by Kwanhian et al. [116]. Among the mutations found were eight cases of single-point mutations in the mature forms of the miR-142 sequence; both miR-142-3p and -5p showcased two cases of single-point mutations in the seed sequence and two outside the seed but within the mature sequence [116]. A total of three cases and six point mutations were found outside the mature -3p and -5p strands of the miR-142 sequence but within the pre-miR-142 sequence [116]. Of the mutations observed (Figure 4), only the mutations in the seed sequence of miR-142-3p resulted in aberrant regulation that had explicable links to cancer [116]. T-to-C and T-to-A mutations at position seven of the miR-142-3p seed sequence caused a loss of function in its ability to inhibit the mRNA of *RAC1*, a Rho GTPase-expressing gene [116,124]. The RAC1 signaling pathway provides an important regulatory mechanism for many modes of tumorigenesis [124]. Activated by CD40 in regulating B lymphocyte proliferation and maturation, *RAC1* overexpression is associated with increased lymphoma cell proliferation in p53-deficient B cells [125,126]. RAC1 can also generate an antiapoptotic complex with BCL-2, which is sometimes activated in DLBCL [127,128]. The potential involvement of loss-of-function mutations in the miR-142-3p seed region may contribute to the development of DLBCL. These mutations could disrupt the inhibitory effect of miR-142 on RAC1 expression, consequently promoting B-cell proliferation while inhibiting apoptosis.

Kuriyama et al. reported an interesting case of a patient with aggressive DLBCL who exhibited chromosomal translocations involving the miR-142 locus and had previously developed autoimmune hemolytic anemia [117]. The t(8;17)(q24;q22) chromosomal translocation resulted in the deletion of miR-142 in the affected allele and the upregulation of *MYC* by flanking it with a miR-142 promoter [117]. Perhaps due to a compensatory reaction after allelic deletion, in vivo studies found miR-142 to be overexpressed, leading to B-cell depletion and altered phenotypes in differentiation [117,129]. Additional investigations are necessary to fully understand the mechanisms involved with these effects, but these findings suggest that deletions in miR-142 can also play a role in DLBCL.

On the other hand, Lawrie et al. demonstrated in a cohort of 80 DLBCL patients’ lymphocyte samples that low expression of miR-142 is indicative of high event-free survival in DLBCL patients who had undergone R-CHOP therapy [130]. This appears to oppose prior findings that showed that the downregulation of miR-142 due to the miR-142-3p seed mutation led to lymphomagenesis in DLBCL. The use of miR-142 as a prognostic biomarker for DLBCL treatment thus necessitates further investigation [130].

#### 3.1.2. Follicular Lymphoma

The second most common subtype of non-Hodgkin lymphoma is follicular lymphoma (FL), which arises from transformed follicular center B cells [118,131]. Despite generally being an indolent cancer, FL exhibits roughly a 30–40% chance of progressing into an aggressive form of FL known as transformed FL. In a small cohort of 12 patients with transformed FL, Bouska et al. found three cases of single-point mutations in miR-142; two of the mutations were in the same location in the miR-142-3p seed identified by Kwanhian et al., and the remaining mutation was in the miR-142-5p seed [118]. Based on the in vitro results in DLBCL from Kwanhian et al., it is reasonable to infer that the same mutation in FL also results in the upregulation of *RAC1*, which could play a role in FL tumorigenesis. Although bioinformatic analysis suggests that the miR-142-5p seed mutation identified could enable targeting of the *CYLD* gene, which negatively regulates NF-κB, the impact on this regulatory pathway was not investigated [118].

Interestingly, Hezaveh et al. also found the same mutation in the miR-142-3p seed in a cohort of 21 patients with FL [115]. Of the two cases of miR-142 mutations and a total of three mutations found, the other two mutations were found outside the seed but within the mature miR-142-3p sequence [115]. Similar to DLBCL, the finding of consistent FL mutations in the -3p seed of miR-142 suggests its significance in lymphoma cancers and is worthy of further investigation in vivo.

#### 3.1.3. Burkitt Lymphoma

Burkitt lymphoma (BL) is another form of B-cell non-Hodgkin lymphoma that is highly aggressive and linked to EBV [132]. Among five BL cell lines, Urbanek-Trzeciak et al. identified a new BL mutation that occurred in the miR-142-3p seed sequence, whose effect mechanistically is still unclear [114].

Additionally, Zhou et al. observed an upregulation of miR-142 in 18 EBV-positive BL patients compared to 16 EBV-negative BL patients [132]. Their study further confirmed that miR-142-5p targets the 3′UTR of *PTEN* mRNAs; miR-142-5p overexpression led to a reduction of *PTEN* mRNA and PTEN protein levels in BL cells [132]. PTEN is a known tumor repressor that acts as a major negative regulator of the PI3K-Akt pathway to induce cell apoptosis [133]. It is, therefore, unsurprising that BL cells treated with mimics of miR-142-5p and an EBV-generated miRNA, EBV-BART6-3p, experienced increased cell viability [132]. These findings are indicative of the role of miR-142-5p upregulation in increasing BL cell survival via *PTEN* [132].

#### 3.1.4. Mucosa-Associated Lymphoid Tissue Lymphoma

Mucosa-associated lymphoid tissue (MALT) lymphoma is another subtype of non-Hodgkin lymphoma that commonly occurs in the stomach and results from chronic inflammation caused by *Helicobacter pylori* (*H. pylori*) infection [134]. Examining 20 MALT lymphoma patient tissue specimens, Saito et al. found overexpression of miR-142-5p compared to nearby non-tumor gastric mucosae [135]. These results were confirmed in vivo, where miR-142-5p was also expressed at higher levels when mice were infected with *H. heilmannii*, the bacterial group to which *H. pylori* belongs [135]. When categorizing patients with MALT lymphoma, it was found that those who developed resistance to *H. pylori* eradication treatment exhibited significantly higher miR-142-5p expression compared to those who later achieved complete remission after the treatment [135]. This illustrates the potential use of miR-142-5p as a prognostic biomarker for patients with MALT lymphoma who have undergone *H. pylori* treatment, primarily through antibiotics. miR-142 was also shown to target the 3′UTR of *TP53INP1* mRNA in vitro, and TP53INP1 protein levels were also lower in both *H. heilmannii*-infected mice and MALT lymphoma patients [135]. P53 notably activates *TP53INP1* for cell cycle arrest and apoptosis [135,136]. This indicates that, ultimately, the upregulation of miR-142 found in MALT lymphomas likely prevents cell death by repressing *TP53INP1* function [135,136]. In another form of cancer (hepatocellular carcinoma), TP53INP1 downregulation has been shown to promote cancer metastasis through the DUSP10/ERK signaling pathway [137].

As for the other mature strand, Fernández et al. demonstrated that miR-142-3p was expressed at higher levels in 10 MALT lymphoma patient tissues than in 3 chronic gastritis tissues [138]. This finding is in line with the results of Gebauer et al., who tested a total of 60 gastritis lesions, 10 of which had a MALT lymphoma score on the Wotherspoon scale [139]. Thus, the upregulation of both mature strands of miR-142 is evidenced in MALT lymphoma, though the molecular effect of miR-142-3p overexpression remains to be investigated.

#### 3.1.5. Mantle-Cell Lymphoma

Mantle-cell lymphoma (MCL) is another aggressive form of non-Hodgkin lymphoma that arises from lymphocytes encompassing a lymphatic nodule [140,141]. MCL is characterized by chromosomal translocation t(11;14)(q13;q32) and abnormal cyclin D1 expression [141]. In a cohort of 30 patients with MCL, Zhao et al. detected downregulation of miR-142-3p and -5p in lymph node specimens (and one spleen specimen) when compared to B lymphocytes of five healthy donors [140]. Moreover, differential expression of miR-142-3p and -5p occurred between poor- and good-prognosis groups of MCL, suggesting its possible use as a prognostic biomarker [140]. In agreement with these findings is a study from Zhang et al., whose analysis of 103 patient MCL tissues also showed a decline in miR-142-5p compared to surrounding non-cancerous tissue [142]. In summary, the downregulation of both mature miR-142 strands is evidenced in MCL and has the potential to be a prognostic biomarker.

#### 3.1.6. Nasal Natural Killer/T-Cell Lymphoma

NKTCL is a relatively uncommon but aggressive subtype of non-Hodgkin lymphoma characterized by injury in the upper respiratory tract and whose development is also linked to EBV [143,144,145]. Motsch et al. reported that miR-142-3p was downregulated in two EBV-positive NKTCL patient tissues and cell lines when compared to five EBV-negative NKTCL patient tissues and CD56+ peripheral blood mononuclear cells (PBMCs), respectively [143]. They confirmed that miR-142-3p targets the 3′UTR of *IL1A* mRNA and that miR-142 expression in cells decreased IL1A protein levels [143]. IL1A is part of the IL-1 signaling pathway that leads to downstream activation of the NF-κB and MAPK signaling pathways for proinflammatory cytokine regulation [146]. Shown to be connected to cell proliferation in NKTCL, IL1A itself is a pro-inflammatory cytokine that can serve as both a para- and autocrine growth factor and an apoptotic inhibitor [147,148]. miR-142-3p has also been shown to inhibit AC9, which consequently reduces cAMP levels for IL1A production [149,150]. Taken together, these findings indicate a potential pathway in which reduced levels of miR-142-3p can both directly and indirectly contribute to EBV-positive NKTCL tumorigenesis by upregulating IL1A levels. Alles et al. supported these findings by demonstrating that co-expression of EBV-encoded RNAs with miR-142-3p further reduced IL1A levels [151].

Adding on to these findings, Chen et al. confirmed in vitro that miR-142-3p targets the 3′UTR of *RICTOR* mRNA. As a key regulator of the PI3K/AKT pathway, RICTOR forms the mTORC2 complex with mTOR to phosphorylate and activate AKT into pAKT [152,153,154]. pAKT overexpression is linked to cancer through the deregulation of various cellular activities, including apoptosis, proliferation, and mobility [155]. The team found that NK cells transfected with miR-142-3p-expressing plasmids resulted in both lower RICTOR and pAKT levels than cells transfected with anti-miR-142-3p-expressing plasmids [153]. Therefore, downregulated miR-142-3p levels could also induce AKT oncogenic pathways via RICTOR in EBV-positive NKTCL. These miR-142 targets, which are most probably involved in NKTCL pathogenesis, are summarized in Table 2.

Choi et al. also found one case of *TP53* point mutation out of five cases of NKTCL [166]. This is potentially relevant, as the study by Motsch et al. showed that the TP53 protein can bind to the promoter region of miR-142-3p in vitro, suggesting a possible regulatory role [143]. Nonetheless, the effect of both wild-type and mutated *TP53* on miR-142 needs further investigation.

#### 3.1.7. Cutaneous T-Cell Lymphoma

Cutaneous T-cell lymphoma (CTCL), the most common primary lymphoma of the skin and another form of non-Hodgkin lymphoma, involves cutaneous-infiltrating cancerous T cells [167,168]. In a study by Sandoval et al., both miR-142-3p and -5p were observed to be overexpressed in the skin biopsies of 32 patients with CTCL compared with five patients with inflammatory dermatosis [169]. These findings are reflected in the results of Shen et al., who reported that miR-142-3p was also expressed at higher levels in the skin biopsies of 50 CTCL patients compared with 25 benign inflammatory dermatosis patients [168]. Manso et al. supported these findings as well by showing that miR-142-3p and -5p were significantly upregulated, specifically in the early-stage mycosis fungoides of CTCL, compared to inflammatory dermatitis in 14 and 15 skin biopsies, respectively [170]. These findings highlight the potential use of upregulated miR-142 levels found in CTCL to separate it from other skin conditions.

#### 3.1.8. Adult T-Cell Leukemia/Lymphoma

Adult T-cell leukemia/lymphoma (ATLL) is an uncommon yet aggressive T-cell neoplasm caused by human T-cell leukemia virus type 1 (HTLV-1) infection [171]. ATLL is another form of non-Hodgkin lymphoma, with one of the worst prognoses [156]. In a study by Bellon et al., seven ATLL patient samples revealed an upregulation of miR-142-3p and -5p compared to the PBMCs of three healthy donors [172]. However, only miR-142-3p upregulation was verifiable in cell lines infected with HTLV-1 when they were compared to normal CD4+ T cells [172]. It was further observed that the upregulation of miR-142 in hematopoietic stem cell progenitors resulted in a significant 30–50% increase in their differentiation into T cells, but not B cells or myeloid cells [172]. In line with this study are the findings from Ruggero et al., who demonstrated that miR-142-3p was expressed nearly three times as often as other miRNAs in T cells after transfection with HTLV-1 [173].

In contrast, a recent study by Ghobadi et al. illustrated that miR-142-3p was significantly downregulated in the blood samples of 8 ATLL patients compared with the blood samples of 10 normal individuals [156]. Noting that Su et al. had previously shown that miR-142-3p can target the 3′UTR of *THBS4* mRNA in vitro [174], Ghobadi et al. measured and observed significantly higher expression of THBS4 protein in ATLL compared to normal samples [156]. They hypothesized that the downregulation of miR-142-3p increased THBS4 expression, which could stimulate tumor angiogenesis, cancer cell migration, and vascular invasion [156,174]. In another type of cancer (prostate cancer), THBS4 was also found to maintain cancer stem cell-like properties via the PI3K/AKT pathway [175]. Nevertheless, due to the contrasting findings, further research is necessary to identify the true direction of aberrant miR-142 expression in ATLL, as different patient demographics or other factors may be at play.

### 3.2. Leukemias

#### 3.2.1. Chronic Lymphocytic Leukemia

Chronic lymphocytic leukemia (CLL) is one of the most commonly reported types of leukemia among adults [176]. CLL arises from neoplastic CD5+ B cells in the blood, bone marrow, or secondary lymphoid tissues [177]. In a cohort of 452 patients with CLL, Puente et al. found a total of eight mutations in miR-142 in five cases [119]. Of the mutations, four were found in the mature miR-142-5p non-seed sequence, one was found in the mature miR-142-3p seed sequence, one was found in the mature miR-142-3p non-seed sequence, and two were found outside the mature miR-142 sequences but within the pre-miR-142 sequence [119]. Rheinbay et al. found the same mutations in a cohort of 90 patients: two in the mature miR-142-5p non-seed sequence and one outside the mature miR-142 sequences but within the pre-miR-142 sequence [120]. miR-142 mutations likely occur at an even lower rate than these studies suggest. Galka-Marciniak et al. found only one sample in 200 CLL cases that exhibited miR-142 mutations. The two mutations identified were again in hotspot areas of the miR-142-5p non-seed and miR-142-3p seed sequences [121]. These mutations are summarized in Table 1. Additional patient cohorts are necessary to determine the role of these mutations in CLL pathogenesis.

Zhu et al. reported that miR-142-5p was upregulated in the B cells of 6 Chinese CLL patients compared to the peripheral B cells of 30 healthy individuals [178]. Contrary to these findings, Zanette et al. found miR-142-5p to be downregulated and miR-142-3p to be upregulated in the peripheral blood samples of nine CLL patients when compared to CD19+ B cells from six healthy donors [179]. Additional biological replicates are required to definitively elucidate the precise mechanism of dysregulation for both miR-142-3p and -5p in CLL and to comprehensively understand their mechanistic effects.

#### 3.2.2. Chronic Myeloid Leukemia

Chronic myeloid leukemia (CML) is a malignant disorder in the hematopoietic stem cell, characterized by the emergence of the *BCR/ABL1* (*BA*) fusion gene from chromosomal translocation t(9;22)(q34;q11) [180]. Chen et al. found that BA expression was followed by reduced miR-142-3p and -5p expression in bone marrow cells from a CML mouse model [157]. Conversely, transfection of miR-142-3p mimics reduced the viability of BA-expressing myeloid cells [157]. A contributing factor for this inverse relationship was shown in vitro to be BA-induced ERK phosphorylation, which partially downregulated miR-142 expression and subsequently disabled miR-142 repression of the anti-apoptotic gene *CIAPIN1* [157]. Of the anti-apoptotic genes tested, only *CIAPIN1*, a downstream regulator of the tyrosine kinase-Ras signaling pathway, showed both reduced mRNA and protein levels in BA-expressing myeloid cells due to miR-142 overexpression [157,158]. These findings highlight an oncogenic regulatory pathway in which CML BA expression reduces levels of both mature miR-142 strands, resulting in *CIAPIN1* inhibition and cell death evasion.

Klümper et al. showed that among 45 treatment-naïve CML patients, miR-142-3p and -5p were downregulated in the peripheral blood and bone marrow samples of those who were later not responsive to imatinib therapy compared to those who were responsive to the therapy [159]. miR-142-5p was further shown to be the only miRNA to also produce the same results in bone marrow samples of the two patient groups, indicating strong potential as a predictive biomarker for CML treatment [159]. Additionally, functional tests found that miR-142-5p could target the 3′UTR of *ABL2*, *MCL1*, *cKIT*, and *SRI* mRNA in vitro [159]. Upregulated in CML progression, ALB2 regulates tumor cell invasiveness, cytoskeleton rearrangement, and matrix metalloproteinase levels [181,182,183,184]. Activated by the GM-CSF, IL-3, and PI3K/AKT pathways, MCL1 is involved in antiapoptotic activity and CML pathogenesis, with reduced levels linked to increased sensitivity to imatinib therapy and overexpression linked to therapy resistance [160,185,186,187]. cKIT, which is involved in the downstream regulation of various pathways, including PI3K, JAK/STAT, Src family kinase, and MAPK, can induce antiapoptotic and pro-proliferative activities [161,188,189,190]. Finally, SRI, which is frequently upregulated in cancer, is associated with tumor growth, the metastatic potential of cancer cells, and the development of resistance to multiple drugs [191,192,193,194,195]. All of these targets suggest possible routes by which miR-142-5p downregulation in imatinib-nonresponsive CML patients could lead to the development of treatment resistance and promote tumorigenesis, conferring more serious conditions than those of imatinib-responsive CML patients.

Interestingly, in an analysis of 11 blood samples of CML patients, Flamant et al. observed a reduction in miR-142-3p expression levels after imatinib treatment [196]. miR-142-3p levels at diagnosis and prior to treatment were also shown to be positively correlated with the Sokal risk score, a prognostic measurement [196]. This indicates that individuals in the chronic phase of CML tend to have increased miR-142-3p expression levels when compared to healthy controls [196].

On the other hand, Ferreira et al., in a study of 75 CML patients and 58 healthy individuals, described lower levels of miR-142-3p expression in patients in advanced CML phases compared to both patients in the chronic phase and healthy individuals [197]. CML patients also experienced increased miR-142-3p expression after treatment with dasatinib [197]. A more extensive investigation is necessary to explain miR-142-3p dysregulation in different CML phases and treatments and the reliability of using miR-142-3p as a prognostic biomarker.

#### 3.2.3. Acute Lymphocytic Leukemia

ALL is a common leukemia among adolescents and young adults, arising from a malignancy in lymphoid progenitor cells and hence affecting both T-cell and B-cell lineages in the bone marrow, blood, and extramedullary sites [198,199,200]. ALL can be broadly categorized into TT-ALL, B-cell ALL (B-ALL), and B-cell precursor ALL (pre-B-ALL) [201]. Dahlhaus et al. illustrated that miR-142 was mostly expressed at higher levels in ALL cell lines and in peripheral blood or bone marrow samples of six patients with ALL, including T-ALL and pre-B-ALL subtypes, compared to CD34+ hematopoietic stem cells of healthy donors [201].

In line with these findings, Lv et al. also found miR-142-3p to be upregulated in peripheral blood samples of 15 T-ALL patients and in T-ALL cell lines compared to the peripheral blood of 10 healthy donors and other cancer/noncancerous cell lines, respectively [162]. Furthermore, high miR-142-3p expression was found to be correlated with short survival times, presenting the possibility of using miR-142 in predicting T-ALL prognosis [162]. miR-142-3p has been demonstrated to target the cAMP/PKA pathway, leading to a decrease in cAMP and PKA activity, which in turn reduces proliferative inhibition in both T-ALL cell lines and primary T-ALL cells. miR-142-3p was also confirmed to target the 3′UTR of *GRα* mRNA in vitro [162]. By targeting the cAMP/PKA pathway and *GRα*, miR-142 promoted glucocorticoid resistance in the same cells [162]. Glucocorticoids continue to serve as an important therapeutic in hematological malignancies by inducing cell death [162]. Huang et al. previously demonstrated that miR-142-3p targets the 3′UTR of *AC9* mRNA, decreasing cAMP production in CD4+ T cells [150]. This points to *AC9* as a direct target of miR-142-3p, affecting the cAMP/PKA pathway as described in T-ALL. miR-142-3p upregulation in T-ALL can, therefore, not only induce proliferation but also present hurdles to common treatments by targeting *AC9* and *GRα*.

miR-142-3p was confirmed by Ju et al. to be drastically overexpressed in six pre-B-ALL patient bone marrow samples compared to six samples from individuals with normal bone marrow [163]. Sakurai et al. also showed that miR-142-3p was more specifically upregulated in pre-B-ALL cell lines that were resistant to treatment with a type of glucocorticoid [164]. *GRα* mRNAs and GRα proteins were both downregulated in dexamethasone-resistant pre-B-ALL cell lines. Linking back to the results of Lv et al., it is highly likely that miR-142-3p also targets *GRα* to induce GR resistance in pre-B-ALL as well [162,164].

On the other hand, Longjohn et al. found in a meta-analysis of miRNAs in 200 B-ALL samples that miR-142-3p was consistently upregulated compared to corresponding healthy controls [202]. However, this was shown to change in the presence of the *MLL-AF4* fusion gene, known to indicate poor prognosis and generated from chromosomal translocation t(4;11)(q21;q23) in ALL [165]. Dou et al. discovered that the primary leukemic blasts of 12 B-ALL patients who were *MLL-AF4+* exhibited lower expression of miR-142-3p than the nucleated bone marrow of healthy individuals [165]. This can be explained by the possible silencing of miR-142 by MLL-AF4, as indicated by the localization of MLL-AF4 at the chromatin regulatory regions of the miR-142 promoter [165]. The shutdown of miR-142-3p is logical considering that miR-142-3p targets *MLL-AF4* in vitro; *MLL-AF4* mRNA and MLL-AF4 protein levels were downregulated in the MLL-AF4+ cell line when transfected with miR-142-3p mimics [165]. Overexpression of miR-142-3p was further demonstrated to inhibit proliferation and induce apoptosis, likely through the effect of MLL-AF4 on the *HOXA9*, *HOXA7*, and *HOXA10* genes, which exhibited concurrent downregulation [165]. To summarize, the presence of MLL-AF4 in B-ALL likely downregulates miR-142-3p to prevent miR-142-3p suppression of MLL-AF4, downstream induction of apoptosis, and inhibition of tumor growth.

#### 3.2.4. Acute Myeloid Leukemia

Acute myeloid leukemia (AML), also one of the most common types of leukemia in adults, arises from the expansion of myeloid precursors with defective differentiation [203]. AML is characterized by chromosomal translocation t(8:21) in core-binding factor AML, generating the chimeric RUNX1–RUNX1T1 protein complex, which restricts maturation of myeloid stem cells [203]. Thol et al. described three cases of mutations in the miR-142 sequence among a cohort of 416 AML patients [122]. miR-142-3p was the most commonly mutated of all miRNAs tested. The three mutations all occurred in the miR-142-3p seed sequence [122]. The TCGA Research Network found five mutations in four samples from a cohort of 200 samples from adults with AML [123]. Here too, all five mutations were located in the seed sequence of miR-142-3p. A pan-cancer analysis conducted by Urbanek-Trzeciak et al. using publicly available TCGA data found two cases and two single-point mutations in the miR-142-3p seed sequence among 149 AML cases (Figure 4) [114]. Although miR-142-3p appears to be a hotspot for mutations in AML, the rate of these mutations is low. In fact, in a pool including 47 AML samples, Galka-Marciniak et al. found no mutations in miR-142 [121].

Investigating the effect of mutations in the miR-142-3p seed sequence, Trissal et al. showed that both the A-to-G mutation at position three and the G-to-C mutation at position six of the miR-142-3p seed caused a loss of function in the ability of miR-142-3p to inhibit the 3′UTR of known targets, such as *RAC1* and *TGFBR1*, in vitro [27]. In addition, recent findings from Kawano et al. also suggest that A-to-G mutations at position 3 of the miR-142-3p resulted in the development of fatal T-cell leukemia in mice [204]. This specific mutation led to a loss of function in the inhibition of the *Rras* and *Rheb* genes but concurrently resulted in a gain of function in inhibiting newly identified target genes such as *Camk1d*, *Bcl2l11*, and *Parp1*. Together, the dysregulation of these genes upregulated the mTORC1 and MYC pathways linked to leukemia differentiation and proliferation.

Although the mutations identified by Trissal et al. were on the miR-142-3p strand, reductions in miR-142-5p and RISC-incorporated miR-142-5p were also observed when compared to wild-type levels [27]. miR-142-3p and -5p were first confirmed to target the 3′UTR of *ASH1L* mRNA, whereas mutated miR-142 at the two sites resulted in a lack of *ASH1L* suppression in vitro [27]. Further confirmation was shown in vivo, where complete miR-142 knockout mice exhibited lower ASH1L protein levels, and isolated bone marrow cells from knockout mice also showed lower *ASH1L* mRNA compared to levels in the wild-type control [27]. Since ASH1L is a positive regulator of *HOX* genes, *HOXA9*/*A10* expression was examined and found to progressively decline along with ASH1L levels for normal murine hematopoietic differentiation [27]. This could explain the upregulation in HOXA9/A10 levels in bone marrow cells and myeloid-committed progenitors under complete miR-142 knockout. The cellular impact could also provide a potential explanation for the aberrant loss of lymphoid potential and maintenance of myeloid potential observed during differentiation [27,205]. Overall, miR-142-3p seed mutations in AML are likely to play a role in the disruption of proper hematopoietic differentiation, which could promote leukemic transformation.

Experiments by Marshall et al. tested the effects of some of the above-described mutations in the miR-142-3p seed—specifically, mutations at positions 3, 5, and 6—which all exhibited loss of function compared to wild-type sequences in vitro [206]. Loss-of-function miR-142s were shown to synergize with the *IDH2*^R140Q^ mutation to cause leukemogenesis both in mice in vivo and in hematopoietic stem/progenitor cells in vitro [206]. Although complete miR-142 knockout in mice resulted in abnormally high levels of myeloblasts, these levels never exceeded 10% of bone marrow CD45+ cells to be considered leukemia development [206]. Similarly, complete miR-142 knockout or the *IDH2*^R140Q^ mutation alone did not cause the development of hematopoietic stem/progenitor cells into myeloid leukemia [206]. The combined effect of loss-of-function miR-142 and *IDH2*^R140Q^ mutations can be potentially explained through the ability of miR-142 to counteract the slight IDH2^R140Q^-caused downregulation of the pro-leukemic *HOXA* cluster, *HOXA5*, *HOXA7*, *HOXA9*, and *HOXA10* [206]. The addition of the *IDH2*^R140Q^ mutation to loss-of-function miR-142 also upregulates *MEIS1* and *PBX3*, which are cofactors necessary for the upregulation of *HOXA9* [206]. Upregulation of the *HOXA* cluster was shown to be at high levels in Mac1+ leukemic cells and was higher in Mac1+ myeloblasts compared to differentiated Mac1+ myeloid cells [206]. Supporting the findings of Trissal et al., Marshall et al. presented evidence that ASHL1 is an intermediate for miR-142 loss of function and *IDH2*^R140Q^ mutational activation of *HOXA* clusters for myeloid progenitor expansion, due to its ability to partially suppress colony formation in these double-mutated cells [206]. Trissal et al. also demonstrated that the addition of loss-of-function miR-142 mutations in *IDH2*^R172K^-mutated mice led to an increase in immature hematopoietic cells with elevated myeloblast levels sufficient to be diagnosed as AML, whereas just *IDH2*^R172K^-mutated mice had insufficient levels for AML diagnosis [27]. Additionally, all patients in the TCGA AML cohort exhibit a mutation in *IDH* [123]. These findings, taken together, demonstrate that the synergy between miR-142 and *IDH* mutations could, in fact, play an important role in AML pathogenesis [123].

In the same ALL study as described above, Dahlhaus et al. also found that miR-142 was mostly expressed at higher levels in AML cell lines and 29 AML patient peripheral blood or bone marrow samples compared to CD34+ hematopoietic stem cells from healthy donors [201]. Among AML patients cytogenetically determined to be in the intermediate-risk group, those who expressed higher miR-142 levels had a significantly better prognosis than those with lower miR-142 levels, indicating the possible use of miR-142 for predicting prognosis in a subset of AML [201].

Contrary to these findings, Wang et al. showed that miR-142-3p was downregulated in the PBMCs across five subtypes of 10 AML patient samples compared to those of six healthy individuals [207]. This could be due to the downregulation of *PPP2R2A* in AML patients compared to normal donors; *PPP2R2A* downregulation also led to reduced miR-142-3p and -5p levels in AML cell lines [208,209]. In addition, an absence of miR-142 mutations was reported by Chen et al. in a cohort of 142 Chinese AML patients [210]. Perhaps differences in miR-142 mutation and dysregulation in AML could be attributed to patients with different backgrounds, but further investigation is necessary.

In samples from six pediatric AML patients, Yuan et al. found that circ-0004136 was the highest expressed circular RNA (circRNA) among the 273 upregulated circRNAs tested [211]. Circ-0004136 was also significantly upregulated in the bone marrow of the six children with AML compared to that of six healthy children. Circ-0004136 was further shown to be capable of acting like a sponge for miR-142, which explains the downregulation of miR-142 and increased proliferation that were observed in an AML cell line [211]. This presents a mechanism involving miR-142 in which leukemogenesis could be potentially promoted in pediatric AML, though confirmation of whether the same miR-142 pathways in AML apply to pediatric AML is needed.

#### 3.2.5. Acute Promyelocytic Leukemia

Acute promyelocytic leukemia (APL) is a subtype of AML characterized by chromosomal translocation t(15:17), which forms a chimeric PML-RARA protein complex that impedes myeloid precursor maturation [203]. All-trans retinoic acid (ATRA) treatment is used in APL to stimulate the differentiation of malignant promyelocytes into neutrophils [212]. Grassilli et al. were able to demonstrate a positive feedback loop between miR-142-3p and both *PU.1* and *Vav1* to carry out the effects of ATRA-induced differentiation in an APL cell line [212]. Vav1 is necessary to recruit the PU.1 transcription factor to the miR-142 promoter at the cis-binding element. When ATRA induces the activation of Vav1 to carry out necessary interactions for myeloid maturation, Vav1 and PU.1 also work together to upregulate miR-142-3p expression, which consequently sustains Vav1 and PU.1 levels [212]. These interactions provide insights into the upregulation of miR-142-3p in APL cell lines during myeloid differentiation induced by both PMA and ATRA, as previously reported by Wang et al. [36]. In their study, miR-142-3p was found to target the 3’UTR of CCNT2, which opposes monocytic differentiation by promoting proliferation. Additionally, miR-142-3p targets the 3’UTR of TAB2 mRNA, and the knockout of TAB2 was associated with granulocytic differentiation [36]. Notably, TAB2 has been shown to be a major activator of the NF-κB pathway and the maintenance of cancer stem cell-like properties in cervical squamous cell carcinoma cells [213]. Reduced miR-142-3p levels, as Wang et al. demonstrated among AML patients, can therefore contribute to APL development by relieving its inhibition of *CCNT2* and *TAB2* targets [36,207]. The upregulation of miR-142 and this molecular feedback mechanism also suggest that PMA and ATRA treatment could together provide sustainable levels of miR-142-3p to induce myeloid differentiation in APL through continued inhibition of *CCNT2* and *TAB2*.

## 4. miR-142 as a Therapeutic Target

Currently, no miRNA-based therapeutic has been approved by the U.S. Food and Drug Administration (FDA). However, many such drugs are undergoing preclinical and clinical trials [16,214]. The most advanced-stage miRNA therapeutic, miravirsen (anti-miR-122 unconjugated) for hepatitis C infection, has demonstrated considerable efficacy in a phase II clinical trial; however, its progress has been somewhat affected by the halting of clinical development of RG-101 (anti-miR-122 GalNAc conjugated), also for hepatitis C infection, due to concerning levels of bilirubin buildup in the blood [16].

Immune-related side effects also present an obstacle to the implementation of miRNA-based therapeutics [215]. This is evidenced in the clinical trial of MRX34 (a miR-34 mimic in a liposome) for the treatment of refractory tumors, which was halted in phase I after severe immune-related side effects led to the death of four patients [216]. Serious adverse events occurred later on in the treatment cycle after the conclusion of daily MRX34 infusions; these events included sepsis, hypoxia, cytokine release syndrome, and hepatic failure, which are symptoms characteristic of immune toxicity [216]. The underlying mechanism of these side effects has yet to be identified. However, the absence of such responses in other oligonucleotide drugs utilizing the same liposome delivery method and the similarity of the side effects to those seen in patients undergoing immune checkpoint inhibitor therapy point to immune-mediated mechanisms driven by MRX34 [216]. Despite these challenges, the potential of miRNA mimics has led to their continued development. Phase I clinical trials have recently been completed for MesomiR-1 (a miR-16 mimic loaded in bacterially derived minicells coated with bispecific panitumumab-based antibodies) with an acceptable safety profile for the treatment of malignant pleural mesothelioma [217,218].

In an effort to circumvent the serious adverse events associated with miRNA overexpression, miRNA inhibitors are being investigated as alternatives with more favorable safety profiles. The use of miRNA inhibitors that differ in their designs, compositions, and properties, which translates to advantages and disadvantages for clinical applications, is thoroughly examined in this review. CRISPR-Cas9 editing is also discussed due to its emergence as a powerful platform for single-base editing of genes encoding miRNA. Still, the future success of miRNA inhibitors and CRISPR-Cas9 editing relies heavily on the ongoing, rapid development of more natural analogs and delivery strategies that seek to transport these therapeutic cargos to the cells of interest, enabling their critical molecular actions.

The evident role of miR-142 dysregulation in lymphomagenesis and leukemogenesis presents opportunities for treatments of hematological malignancies focused on correcting miR-142 mutations and reversing aberrant miR-142 expression levels. For example, miR-142 mutations in DLBCL, FL, BL, CLL, and AML can be revised using CRISPR-Cas9 DNA editing of the miR-142 gene. On the other hand, miR-142 can be overexpressed with mimics to counteract both miR-142 loss-of-function mutations and downregulated expression in DLBCL, FL, NKTCL (EBV+), ATLL, CML, B-ALL, AML, and APL. miR-142 inhibition can be performed via anti-miRNA oligonucleotides (AMOs), miRNA sponges, and circRNAs to counteract miR-142 upregulation in BL (EBV+), MALT (*H. pylori* resistant), T-ALL, and pre-B-ALL. Understanding the multifaceted role of miR-142 in hematopoietic physiology also necessitates consideration of anticipated side effects associated with miR-142-based therapies and methods to minimize them.

### 4.1. Anti-miRNA Oligonucleotides

AMOs are a class of synthetic, cholesterol-conjugated RNA oligonucleotides (about 23 nucleotides), termed “antagomiRs,” which bind to a natural miRNA with full complementarity [219,220]. Chemical modifications of single-stranded, anti-sense antagomiRs are necessary to improve their stability by preventing ribonuclease degradation and phagocytosis by the reticuloendothelial system during in vivo delivery [221]. Phosphorothioates (PSs) are a common backbone modification in which sulfur is swapped with nonbridging oxygen in the phosphate group, maintaining an anionic linkage [222]. A modified PS backbone provides nuclease stability to the natural phosphodiester internucleotidic linkage that is sensitive to endo- and exonucleases in the serum [223,224]. PS modification also increases the rate at which oligonucleotides bind to proteins in the plasma, lessening their renal clearance and increasing their chance for cellular uptake [225]. Other backbone modifications, such as boranophosphate, 3′-methylenephosphonate (3′-MEP), and 5′-MEP, provide anionic internucleotide linkage, while guanidinopropyl phosphoramidate, deoxynucleic guanidine (DNG), and deoxynucleic S-methylthiourea (DNmt) provide cationic linkage, and methylphosphonate provides neutral linkage [224]. These modifications not only contribute to stability but also enhance the binding affinity of the oligonucleotide to its complementary target [224].

In addition to primary PS modification, second-generation chemical modifications of the sugar moiety in the 2′ position of the ribonucleotide can further bring beneficial properties to AMOs, such as pharmacokinetics, toxicology, and binding affinity [224,226]. For example, the 2′-fluoro modification can aid in AMO preservation and silencing while reducing immunogenicity [227]. The 2′-O-methyl (2′-O-Me) modification confers nuclease resistance and reduces immunogenicity and toxicity by minimizing nonspecific protein binding [228]. The 2′-O-methoxyethyl (2′-O-MOE) modification augments these favorable properties [228]. A miR-142a-5p antagomiR incorporating both PS and 2′O-MOE modifications successfully downregulated osteoblast differentiation by derepressing nuclear factor IA (NFIA), a site-specific transcription factor, in stromal cell line ST2 and pre-osteoblastic cell line MC3T3-E1 in vitro [229].

Lock nucleic acids (LNAs) are characterized by sugar modification that exhibits advantageous binding affinity, high specificity, and potency; this is exemplified by the inhibition of an entire miRNA family that shares a seed sequence with seven to eight nucleotides of “tiny LNAs” (LNA-antimiRs) with concurrent high nuclease stability and reduced immunogenicity [230,231,232]. LNAs represent a promising technology for therapeutic translation and are currently in clinical trials [233]. For instance, cobomarsen (MRG-106), currently in a trial, utilizes an LNA design for the treatment of the mycosis fungoides stage of CTCL by inhibiting miR-155 [234]. In the research setting, intraperitoneal injection of LNA-anti-miR-142-3p targeting dendritic cells reduces endotoxin-induced mortality in the in vivo model of Gram-negative sepsis [68]. Intravitreal injection of LNA-anti-miR-142-3p diminishes microglia immune cell inflammation and VEGF-A-induced angiogenesis in a mouse model of age-related macular degeneration [235]. Intravenous injection of mesenchymal stem cell-derived exosomes loaded with LNA-anti-miR-142-3p effectively delivers the complex to subcutaneously implanted 4T1 tumors, reduces tumor growth, and increases survival in female BALB/c mice [236]. On the other hand, LNA-anti-miR-142-5p administered intraperitoneally improves inflammation in experimental colitis via the IL10RA pathway, decreases weight loss, and increases the overall survival of mice [237]. LNA-anti-miR-142-5p intravenous treatment abrogates CCL4-induced liver fibrosis and bleomycin-induced lung fibrosis perpetuated by profibrogenic macrophages with IL-4- and IL-13-driven miR-142-5p induction and TGFβ1 production [238].

In addition to separate phosphate and sugar modifications, peptide nucleic acids (PNAs) and phosphorodiamidate morpholino oligonucleotides (PMOs) can replace the entire sugar-phosphate backbone [239]. PMOs are uncharged phosphorodiamidate linkage-formed morpholine rings that are highly nuclease-stable [239]. PMOplus oligonucleotides are much like PMOs but are positively charged and contain a piperazine moiety in their phosphorodiamidate backbone [224]. The use of a miR-142-3p PMO enables the interrogation of miR-142-3p and *cdh5* gene regulation in a Zebrafish model of vascular development and enables the identification of miR-142-3p’s predominant role in controlling hematopoietic stem cell lineage programming in a Xenopus model of ontogeny [240,241]. PNAs are uncharged oligonucleotides (13-18 nucleotides) with a backbone made of peptide bonds linking N-(2-aminoethyl)glycine units [241]. This structure increases stability, nuclease resistance, and binding affinity [242]. Nonetheless, PNAs are associated with low bioavailability and solubility, which restricts their application and prompts a need for more analog development [243]. The use of miR-142-3p PNA resolves the wing-versus-leg type-specific chondrogenic effect of TGFβ3 via induction of miR-142-3p and a membrane-anchored protease, ADAM, which mediates cell-to-cell communication and migration [244].

Despite the chemical advancements, the disadvantages of AMOs in clinical translation are becoming more evident. First, AMOs require high and frequent dosages to sustain a therapeutic level of AMOs for effective miRNA inhibition [245,246]. The sophistication of current delivery systems falls short of overcoming this obstacle. Second, following successful transfection, AMOs trapped in intracellular vesicles without endosomal escape to the cytoplasm can neither participate in miRNA inhibition nor bypass lysosomal degradation [247]. Third, the hurdle of immunogenicity still hampers these unnatural synthetic AMOs [16]. Single-stranded RNAs recognized as foreign antigens trigger pathogen-associated molecular pattern (PAMP) receptors, including Toll-like receptors (TLRs), which act as part of the viral defense mechanism in our immune system [248]. RNAs are more immunogenic if they are longer than 20 ribonucleotides and single-stranded rather than double-stranded, with both resulting in TLR activation [249,250]. The use of tiny LNAs could be a solution to overcome immunogenicity, but smaller nucleotides increase the potential for off-target effects [231].

### 4.2. miRNA Sponges

The shortcomings of AMOs have prompted the development of new synthetic miRNA inhibitors. The miRNA sponge is an artificial RNA transcript transcribed from a DNA-encoded plasmid consisting of an RNA polymerase (Pol) II or III promotor, a reporter gene, and multiple miRNA-recognizing elements (MREs) in the 3′UTR to bind to complete or partial complementary miRNA targets [251].

In contrast to AMOs, which lack inducible and durable miRNA suppression, an miRNA sponge can be expressed from an inducible and tissue-specific promotor in a target cell of interest, resulting in more prolonged suppression against a specific miRNA, multiple different miRNAs, or an entire miRNA family [252,253,254,255]. Both RNA Pol II/III promoters are capable of expressing an miRNA sponge [256]. Unlike RNA Pol III-transcribed inhibitors, RNA Pol II transcripts have a 5′ cap structure and 3′ poly(A) tail, which aid its stabilization [256]. However, miRNA sponges derived from Pol II have higher tissue specificity. The reporter gene indicates miRNA sponge expression. When miRNAs bind to the sponge, expression of the reporter gene is interrupted, demonstrating that the sponge is soaking up miRNAs. Evidence suggests that placing the sponge in the 3′UTR region of a coding reporting gene provides more efficient miRNA sequestration compared to placing it downstream of a non-coding transcript [251]. Moreover, the absence of conventional secondary RNA structures in the 3′ UTR region facilitates miRNA binding to the sponge [257].

A miRNA sponge typically contains 4–16 MRE sequences serving as miRNA-binding sites to sequester miRNAs [258]. While more MREs could increase the potential for more miRNA sequestration, over-extending MREs increases the chance for undesired RNA recombination and degradation [259]. Spacers (typically 4–6 nucleotides) divide each MRE to maximize the number of bound miRNAs and minimize the possibility of RNA secondary structure formation [260]. Just like general miRNA regulation of mRNAs, the perfect complementary base pairing of target miRNAs to the MREs induces the RISC/Ago2 complex to cleave and degrade the sponge [261]. Partial mismatch base pairing of target miRNAs to a bulge structure in the middle section of the MREs decreases endonucleolytic cleavage, improves binding affinity to the sponge, and thus increases miRNA inhibition [262]. However, increasing the number of binding sites creates less stable bulged sponges, increasing the likelihood of deletions and mutations [263].

Reciprocal miR-142-3p and miR-122 miRNA sponges, each with four MREs built on a Cre recombinase (Cre)-loxP system delivered intravenously by an adenoviral vector, efficiently suppress residual hepatic transduction and transgene expression while maintaining expression in the target spleen organ for immunotherapy gene manipulation [264]. Ex vivo lentiviral transduction of a miR-142-3p miRNA sponge with multiple MREs into CD11b− bone marrow cells downregulates miR-142-3p expression and promotes macrophage CD11b+ marker expression and the acquisition of a tumor suppressive phenotype in vivo. In comparison, enforced lentiviral expression of miR-142-3p prevents tumor-driven macrophage differentiation and increases effective adoptive T-cell therapy [265]. Finally, intraperitoneal injection of an adeno-associated virus (AAV)-packaged miR-142-3p miRNA sponge inactivates miR-142-3p expression in beta cells of NOD mice, thereby reducing the incidence of Type 1 diabetes [106].

The “Tough Decoy” is a specialized miRNA sponge with a hairpin RNA structure consisting of two bulged miRNA-binding sites directly facing one another, surrounded by two stems and a small loop on top [266]. The bulge prevents endonucleolytic cleavage, and the double-stranded structure impairs RISC-mediated destabilization of the miRNA sponge [266]. Both RNA Pol II and III have been used to generate the Tough Decoy, with their own sets of advantages, as described previously [259,266]. Interestingly, a Tough Decoy with only two miRNA-targeting sites shows miRNA suppressive activity comparable to that of a conventional bulged sponge with eight miRNA-targeting sites. Moreover, Tough Decoy is less susceptible to non-specific targeting by endogenous miRNAs [259]. For example, overexpression of a miR-142-3p Tough Decoy downstream of a strong U6 promotor in a lentiviral vector is effective as a sponge for even the highly abundant, intracellular miR-142-3p in loss-of-function studies in transduced cells. Similarly, a large-scale miRNA decoy library can be used to perform high-throughput, global assessment of pooled loss-of-function studies within a cell [267]. These are valuable research tools for investigating miRNA biology for diagnostic and therapeutic developments.

miRNA sponges have shown great utility in medical research but have not yet been applied directly in the clinic. This is because DNA-based plasmids that express miRNA sponges often require a viral delivery vehicle for effective transport into the cell for permanent miRNA suppression [258]. Pre-existing or acquired immunity, replication capability, genome integration, and latent oncogene activation are all legitimate concerns for using viral delivery. We will address these issues in greater detail in Section 4.5, Delivery.

### 4.3. Circular RNAs

CircRNAs in eukaryotic cells were discovered more than 30 years ago as possible viral-derived products [268]. Only within the last 10 years have two human endogenous circRNAs anti-sense to the Cdr1 transcript and sex-determining region Y clearly demonstrated their roles as natural and highly stable miRNA sponges [269,270]. Endogenous circRNAs are formed by non-canonical backsplicing of precursor messenger RNA (pre-mRNA) when a 5′splice donor disrupts a 3′ splice site, resulting in the formation of a phosphodiester bond [271]. The 5′ UTR of the coding exon serves as the origin of these circRNAs [272]. The production of circRNAs strongly depends on introns with long inverted repeats [273]. ALU reverse complementary repeats also predict which exons are likely to circularize [274]. After backsplicing, circRNAs undergo nuclear export into the cytoplasm, where they sponge miRNAs or release them into the extracellular space via exocytosis [271]. These natural circRNAs often have multiple MREs to maximize the miRNA sponge effect [270,275].

CircRNAs have been demonstrated to act as miR-142-3p sponges. Natural endogenous circCUL2 as a tumor suppressor inhibits gastric cancer cell proliferation, migration, and invasion by sponging miR-142-3p in subcutaneous tumor xenograft models in nude mice. CircCUL2 also regulates cisplatin chemosensitivity through miR-142-3p/ROCK2-mediated autophagy activation [276]. Conversely, endogenous circZNF609 as a tumor promoter increases lung cancer cell proliferation and migration by sponging miR-142-3p in a subcutaneous A549 xenograft model in BALB/c athymic nude mice. An upstream FUS RNA-binding protein induces and upregulates circZNF609 expression by binding to ZNF609 pre-mRNA in lung cancer [277]. In a HCT116 colorectal metastasis model in nude mice, cancer-derived exosomal circPACRGL enhances colorectal cancer cell proliferation, migration, and invasion by sponging miR-142-3p, thereby increasing TGFβ1 expression. As a pleiotropic cytokine, TGFβ1 is intricately involved in tumor initiation, progression, and metastasis.

miR-142-5p sponging by endogenous circIGHG, a human immunoglobulin heavy chain G-derived circRNA, enhances epithelial–mesenchymal transition by derepressing IGF2BP3 expression in a subcutaneous xenograft model of oral squamous cell carcinoma [278]. Epithelial-mesenchymal transition is a critical cell-conversion step toward cancer metastasis [279]. miR-142-5p sponging by endogenous circTMEM87A increases cell growth, migration, invasion, and anti-apoptosis by derepressing ULK1 autophagy kinase in a subcutaneous gastric cancer xenograft model in nude mice [280]. Interestingly, autophagy has a dual role in carcinogenesis—as a tumor suppressor before cancer initiation but a tumor promoter during cancer progression [281]. Finally, miR-142-5p sponging by doxorubicin drug-induced endogenous cirRNA_0004674 promotes osteosarcoma progression and chemoresistance by derepressing anti-apoptotic MCL1 protein expression in an orthotopic tumor xenograft model in nude mice [282].

In contrast to endogenous circRNAs, synthetic circRNAs are artificial miRNA sponges derived from linear intron-containing mRNA precursors by enzymatic digestion and permuted intron–exon (PIE) methods. Enzymatic digestion is an in vitro method in which the specific linear RNA strands can circularize by 5′–3′ end-to-end ligation using T4 bacteriophage RNA ligase [283]. In the PIE method, circRNAs can be generated in vitro and in vivo through self-backsplicing in the group I intron derived from the T4 thymidylate synthase (td) gene or solely in vitro by the Anabaena pre-tRNA(Leu) gene [284,285]. However, the selection of the group I intron influences circRNA yield [286]. For instance, a significant increase in circRNA formation was observed through the luciferase activity in a PIE luciferase construct with Td introns compared to one with Anabaena pre-tRNA introns [286]. Multiple MREs built in tandem within the engineered circRNAs increase miRNA sponge capacity, mimicking the function of endogenous circRNAs [287]. Theoretically, replacing miRNA-binding sites with a Tough Decoy sponge may enhance the efficacy of the circRNA sponge, but more functional validation is necessary [288].

Synthetic circRNA sponges also mirror their natural counterparts, lack a 5′ cap and a 3′ poly(A) tail, and are more resistant to miRNA-mediated RNA destabilization via deadenylation and exonucleolytic degradation [289]. CircRNAs circumvent the challenges of the high dosage dependency of AMOs required for effective treatment and the inability to control the quantity of miRNA sponges synthesized in vivo from plasmids. CircRNA sponges also degrade much more slowly in comparison to linear RNAs and accumulate to higher levels in cells to derepress miRNA targets [290]. Production of circRNA sponges is straightforward, as nuclease digestion helps to remove RNA impurities from the synthesis, and no additional chemical modifications are required [291]. Unlike AMO-related immunogenicity, endogenous circRNAs are inherently non-immunogenic [292]. The immunogenicity of synthetic circRNA sponges depends on major contextual factors, such as intron identity, enzymatic generation of circRNAs, N6-methyladenosine (M6A) RNA nucleoside modification, and purification stringency. For example, utilization of a PIE construct with the T4 group I intron or splint ligation with T4 DNA ligase promotes an immune response [293]. Lack of m6A nucleoside modifications in the PIE-derived circRNA increases immune activation, whereas the incorporation of M6A-modified nucleotides when the construct is undergoing in vitro transcription can ultimately minimize immune recognition [294]. Post-synthesis purification strategies such as size-exclusion high-performance liquid chromatography, phosphatase dephosphorylation, and RNase R degradation of linear RNAs can also be used to decrease innate immune triggers [295]. As the circRNA field continues to evolve, researchers are rapidly gaining insights into the subtle elements contributing to circRNA immunogenicity [296].

Therapeutic application of synthetic circRNA sponges for malignancies is in its infancy. Recent investigations focus on circRNA sponge therapy for solid tumors and have not yet applied to hematologic malignancies [297,298]. The circRNA sponges in these solid tumor models utilize in vivo circRNA production with a PIE-like circularization process from a transcribed plasmid template and are delivered with a lentiviral vehicle, thus being useful only for feasibility testing and proof-of-concept studies. However, the pace of technical advancements has accelerated in recent years for in vitro circRNA manufacturing and quality controls for clinical translation. Among the current therapeutic approaches to rectifying miRNA dysregulation, synthetic circRNA sponges have the most potential to achieve this goal.

### 4.4. CRISPR-Cas9 for DNA Knockout and Editing

Clustered regularly interspaced short palindromic repeats (CRISPR)-Cas9 is a powerful genomic-editing tool that uses guide RNA to recognize 20 nucleotide sequences and the nuclease activity of Cas9 to make double-strand breaks at the specified location [299]. Utilizing non-homologous end joining and homology-dependent repair, CRISPR-Cas9 enables gene modification at desired targets in the genome [299]. CRISPR-Cas9 DNA has already been applied to generate deletion mutants of miR-142-3p in vitro and to delete multiple miR-142-3p recognition elements in the 3′ UTR of the *Cdkn1b* gene in vivo [300,301]. Still, CRISPR-Cas9 has disadvantages. Cas9 un-specifically cleaves many off-target sequences throughout the genome and may cause deletions in the intended target DNA sequence, including large mono-allelic genome deletions and subsequent loss of heterozygosity [302,303]. Moreover, immunity to Cas9 bacterial protein from pre-existing and acquired exposures is being recognized as an obstacle to persistent genome editing [304]. Finally, delivery of the RNA editing cargo to the specific cells of interest currently stands as the biggest challenge to in vivo validation studies and clinical translation for hematologic malignancies [305].

### 4.5. Delivery

The design of an anti-miRNA therapy dictates the delivery system best suited to transport the therapeutic cargo into the cell to trigger a biological response. AMOs, miRNA sponges, circRNA sponges, and CRISPR-based RNA editing each have a different chemical composition, charge, and size that must be packaged inside a delivery vehicle, transported in the bloodstream, uptaken by the cell, and released into the correct subcellular compartment to initiate their molecular action. For AMOs and circRNA sponges, the cytoplasm is the transport destination and the primary site of anti-miRNA actions. For plasmid-based miRNA sponges, the nucleus is the destination site, but the cytoplasm is the site of action for the transcribed mRNA. For CRISPR-based RNA editing, the action can be cytoplasmic or nuclear, depending on the targeted transcript. In essence, the delivery vehicles must be suited to their cargo’s functions.

Viral delivery is currently the most suitable and popular nuclear transport vehicle for plasmid-based miRNA sponges [251,258]. Viral vectors are also highly efficient at transducing most human cells [306]. Recombinant viral vectors can consist of integrating retroviruses and lentiviruses or non-integrating adenoviruses and adeno-associated viruses (AAVs) [307]. Non-integrating viruses and less immunogenic AAVs have the best opportunity for in vivo clinical applications [308]. AAV vectors also allow for tissue selectivity based on AAV serotypes and stable or transient gene expression based on cell-specific promotors [308]. AAVs have a smaller cloning capacity (~4.7 kb) compared to other viral vectors, but this is not a concern for shorter-length miRNA sponges [251,309]. Like adenoviruses, the main disadvantage of AAVs is their pre-existing immunity against the wild-type virus, resulting in the rejection of the delivery vehicle [310]. A study of international cohorts of healthy donors found that the prevalence of pre-existing immunity as measured by anti-AAV1 neutralizing antibodies was around 27% for AAV1, 47–74% for AAV2, and 32–63% for AAV8 [310]. To counter the effect of these neutralizing antibodies, the AAV can be made less immunogenic through PEGylation chemical modifications, immunosuppressive drugs, plasmapheresis, higher doses, and balloon catheters [311,312]. These persistent developmental efforts have led to the FDA approval of two AAV-based therapies to treat a form of congenital blindness (AAV2-based voretigene neparvovec (Luxturna) in 2017) and spinal muscular atrophy (AAV9-based onasemnogene abeparvovec (Zolgensma) in 2019) [313].

Integrating viral vectors such as retroviruses and lentiviruses are predominantly utilized in experimental models and research [251]. Retroviral vectors transduce only dividing cells, while lentiviral vectors transduce both non-dividing and dividing cells [251]. Lentiviral vectors confer more flexibility in usage and has undergone three generations of vector safety improvement [314]. It successfully gained FDA approval in 2017 for chimeric antigen receptor (CAR) T-cell therapies, axicabtagene ciloleucel (Yescarta), tisagenlecleucel (Kymriah), and most recently (2021), brexucabtagene autoleucel (Tecartus), but only in the ex vivo setting [315]. For in vivo usage, the main concerns for integrating viruses are insertion mutations, oncogenic activations, and malignant cell transformations. An example is the occurrence of T-fcell leukemia-like disease following gene therapy in patients with SCID-X1 who received retroviral vectors [62]. Integrating viral vectors are highly immunogenic and are hampered by acquired immunity from repeated injections [316].

An ideal delivery method should be non-toxic, non-immunogenic, serum-stable, tissue- and cell-type-specific, high in cellular uptake, efficient in endosomal release, and biodegradable. To overcome the current limitations of viral vectors, nonviral vectors are becoming desirable substitutes for safer, more stable, more specific, and cheaper delivery, particularly for single-stranded, chemically engineered AMOs and in vitro synthesized circRNA sponges [317]. The two main types of non-viral vectors are polymer- and lipid-based delivery systems.

The gold standard polymer-based delivery method is linear or branched polyethyleneimine (PEI) with multiple positively charged amines enabling endosomal release through a “proton sponge effect” [317]. This process is characterized by an alteration of the acidic vesicle osmolarity, which causes endosomal swelling, rupturing, and the release of RNA molecules [317]. PEI is highly effective in vitro, but a lack of degradability and toxicity limits its in vivo usage, particularly after repeated administrations [318]. Polyethylene glycol (PEG) and poly-L-lysine (PLL) can lessen toxicity compared with PEI, but safety issues still ensue [319,320]. Poly(lactide-co-glycolide) (PLGA) is a polymer composed of poly(glycolic acid) and poly(lactic acid) (PLA), which received FDA approval in 1989 [321]. PLGA has excellent biocompatibility, biodegradability, and capability to envelop both hydrophobic and hydrophilic cargo types for protection [322]. In vivo safety studies of animals treated with PLGA nanoparticles containing miRNA-99a mimics targeting hepatocellular carcinoma have not reported issues with toxicity, blood chemistry alterations, or drastic changes in body weight [323]. The main drawbacks of PLGA’s use in miRNA delivery are hydrophobicity and rapid opsonization during circulation; however, PEGylation can increase the circulation time of PLGA nanoparticles [322,324]. An interesting application of PLGA delivery, a miniaturized biodegradable polymer matrix (LODER, LOcal Drug EluteR) for slow release of an siRNA drug after local injection, is currently being tested for pancreatic cancer treatment in a phase II clinical trial [325,326].

Lipid-based delivery includes cationic liposomes and more advanced ionizable cationic lipid nanoparticles (LNPs) [327]. Liposomes composed of a lipid bilayer allow the oligonucleotide to encapsulate in its hydrophilic aqueous core or complex with its hydrophobic bilayer [324,327]. To maximize oligonucleotide loading, cationic lipids are often more preferable; 1,2-dioleoyl-3-trimethylammonium propane (DOTAP) and N-(1-(2,3-dioleyloxy)propyl)-N,N,N-trimethylammonium chloride (DOTMA), with permanent positive charges, can better associate with the negatively charged oligonucleotides [324,328]. However, these permanently charged cationic lipids are highly immunogenic and tend to have an affinity for serum proteins and blood cells that are negatively charged, resulting in low delivery efficiency and higher toxicity during in vivo applications [324,329]. Successful development of ionizable cationic lipids, which remain positively charged in acidic environments during ex vivo complex formation with negatively charged oligonucleotides and in vivo endosomal escape but are almost neutral at physiological pH once inside the bloodstream, led to the FDA approval of DLin-MC3-DMA delivery for patisiran (Onpattro) siRNA-based therapy in 2018 [330]. The success of MC3 has facilitated the development of COVID-19 vaccine delivery of BioNTech ALC-0315 and Moderna SM-102 cationic lipids, which are not only ionizable but also biodegradable [331]. To mimic natural physiologic lipids, LNPs have 1,2-distearoyl-sn-glycero-3 phosphocholine (DSPC) and cholesterol as “helper lipids” in their formulations to strengthen stability and PEG to reduce aggregation [332]. To achieve next-generation tissue- and cell-specific delivery, ligand conjugations of LNPs using polysaccharides, peptides, proteins, and antibodies are effective strategies to enhance active targeting, but inherent immunogenicity related to the ligand’s composition, size, and biological humanization must also be considered [333,334,335,336]. Based on the commercial success of biodegradable, ionizable cationic lipids, the future looks bright for their application in miRNA therapeutics.

## 5. Concluding Remarks and Outlook

miRNAs are essential regulatory molecules, and their dysregulation is frequently associated with cancer. miR-142 is a unique miRNA that is enriched in hematopoietic cells for proper gene control and transported by exosomes for various physiological functions. The essential role of miR-142 in the maintenance of hematopoietic homeostasis and response explains the occurrence of miR-142 mutations and aberrant expression in hematological malignancies. Mechanistically, many of these irregularities have been shown to be linked to lymphomagenesis and leukemogenesis. To counter the carcinogenic effects of miR-142 dysregulation in leukemias and lymphomas, therapeutic interventions utilizing miRNA inhibition and editing platforms present promising solutions, although both still require continued testing and improvement. Depending on the malignancy, further bioinformatic analysis and in vivo animal models may also be necessary to concretely determine the effect of genetic abnormalities on miR-142 function and the efficacy and safety of miR-142-based treatment.

In addition, mutations occurring in pri-miRNA processing sites, such as the CNNC-, UG-, and UGU/GUG/UGUG motifs, can impact miRNA biogenesis and function. A recent study identified SRSF3 as the most enriched protein interacting with pri-miR-142 and described the expression of a mutant pri-miR-142 exhibiting a modified CNNC motif [337]. SRSF3 is a splicing factor that binds to the CNNC motif and is involved in the alternative processing of most pri-miRNAs [337]. Alternative processing via SRSF3 was shown to be reduced in mutant pri-miR-142 CNNC motifs compared to that of wild-type motifs [337]. These new findings demonstrate that CNNC motif mutations could impact the biogenesis of mature miR-142 strands. Future research in pri-miR-142 CNNC-, UG-, and UGU/GUG/UGUG motif mutations in the context of hematological malignancies could enable a better understanding of the sources of extensive miR-142 dysregulation and function.

Studies have shown that epigenetic modifications can serve as a cause of miRNA dysregulation and subsequent malignant transformation of otherwise healthy cells. miRNAs are epigenetically controlled through genomic methylation and chromatin modifications at regions of miRNA loci but also contribute to epigenetic control by targeting epigenetic modifiers [338]. This interaction forms an epigenetic loop involving miRNA. Epigenetic control of miR-142 has been identified as a contributing factor to malignant phenotypes in cancers such as glioblastoma, nasopharyngeal carcinoma, and breast cancer [339,340,341,342]. Investigating the effect of miR-142 on epigenetic enzymes and the role of miR-142 in epigenetic control loops, specifically in hematological malignancies, could not only deepen our understanding of miR-142 mechanisms in lymphomagenesis and leukemogenesis but also provide valuable clinical targets.

## Figures and Tables

**Figure 1 cells-13-00084-f001:**
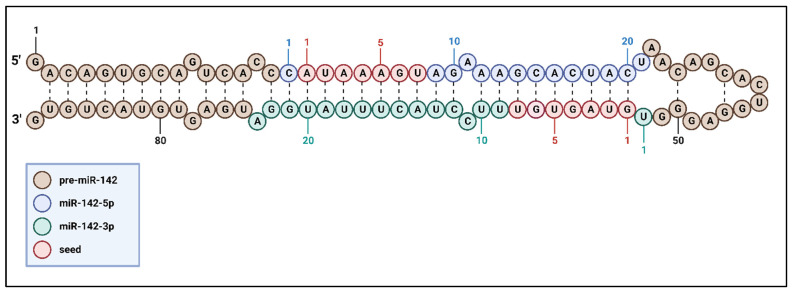
Components and relative positions of the secondary stem-loop structure of the pre-miR-142 sequence. The circles represent nucleotides; the circles’ colors correspond to the component of pre-miR-142. Each colored number corresponds to the nucleotide position relative to the specific component of pre-miR-142.

**Figure 2 cells-13-00084-f002:**
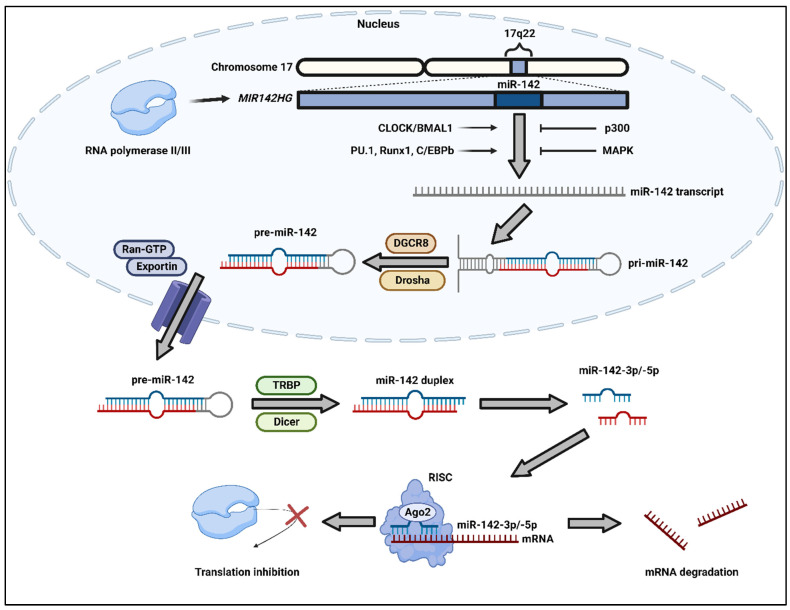
Biogenesis and regulation of miR-142. The transcription of the *MIR142HG* gene via RNA polymerase II/III is regulated by proteins, including CLOCK/BMAL1, PU.1, Runx1, and C/EBP transcriptional promotion and p300 and MAPK transcriptional repression, dependent on the cell type or larger function of the organ system. The transcript forms the pri-miR-142 structure, which is subsequently cleaved by DGCR8 and Drosha to form pre-miR-142. This is released into the cytoplasm through the facilitation of exportin-5/Ran-GTP and processed via TRBP and Dicer to generate the mature miR-142 duplex. Either one or both mature miR-142-3p and -5p strands can be incorporated into the RNA-induced silencing complex (RISC) to enact post-transcriptional regulation by targeting mRNAs. The binding of the miRNA RISC complex leads to either mRNA degradation or translational inhibition.

**Figure 3 cells-13-00084-f003:**
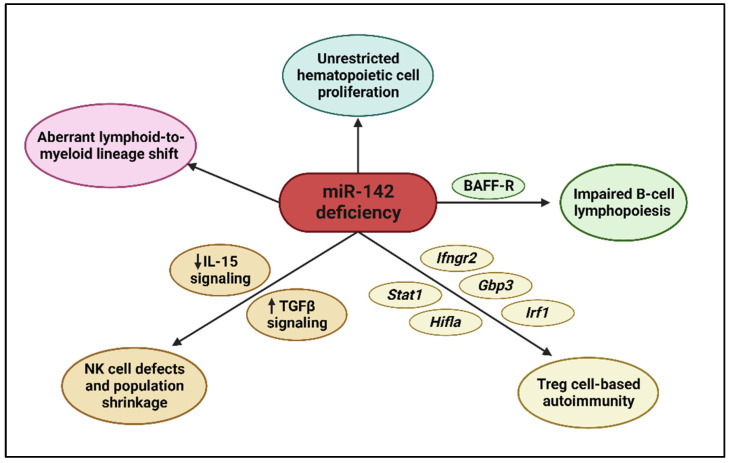
The effects of miR-142 deficiency on hematopoiesis and hematopoietic cells, including B cells, T cells, natural killer (NK) cells, and myeloid cells. Affected targets and downregulated or upregulated signaling pathways are shown.

**Figure 4 cells-13-00084-f004:**
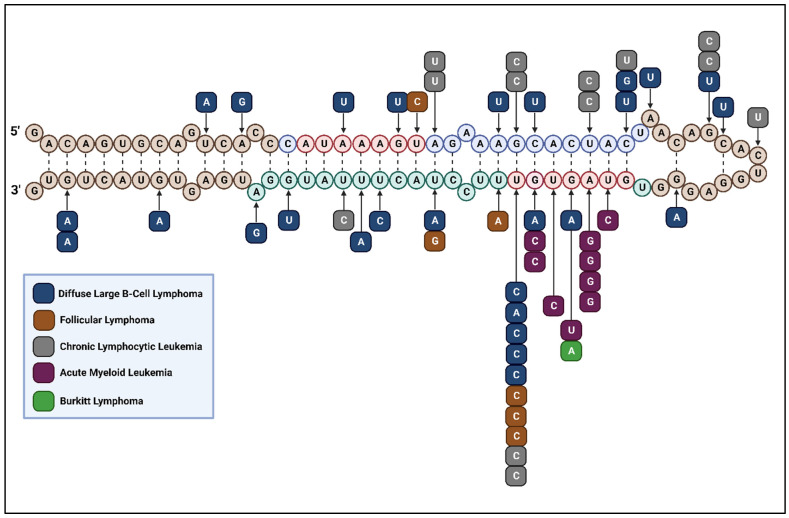
Mutations identified within the secondary stem-loop structure of the pre-miR-142 sequence. The circles represent nucleotides. Point mutations found in hematological malignancies described in this article are indicated by arrows with nucleotides encased in squares; the squares’ colors correspond to the cancer type. The components of pre-miR-142, which include the mature 3p strand, 5p strand, and seed sequence, are as depicted in Figure 1 (the miR-142-5p sequence is represented by nucleotides encased in blue circles, the miR-142-3p sequence is represented by nucleotides encased in green circles, and the seed sequences in each mature sequence are represented by nucleotides in red circles).

**Table 1 cells-13-00084-t001:** Locations of mutations identified in miR-142 in each cancer type and the corresponding frequency of each mutation within the cohort examined.

Cancer Type	Cancer Subtype	Mutation Position	Position Reference	Occurrence of Mutation in Each Position	Reference
Lymphoma	DLBCL	Positions 13, 21	miR-142-3p	1/37 cases	[114]
		Position 6	miR-142-3p seed		
		Positions 80, 85	pre-miR-142		
		Position 20	miR-142-5p	2/37 cases	
		Position 7	miR-142-3p seed	3/37 cases	
		Position 4	miR-142-3p seed	1/19 cases	[115]
		Position 17	miR-142-3p		
		Position 50	pre-miR-142		
		Positions 3, 6	miR-142-5p seed	1/56 cases	[116]
		Positions 13, 15	miR-142-5p		
		Positions 16, 23	miR-142-3p		
		Positions 11, 13, 37, 41, 42, 85	pre-miR-142		
		Position 7	miR-142-3p seed	2/56 cases	
		t(8;17)(q24;q22)	pre-miR-142	1/1 case	[117]
	FL	Position 7	miR-142-5p seed	1/12 aggressive cases	[118]
		Position 7	miR-142-3p seed	2/12 aggressive cases	
		Position 7	miR-142-3p seed	1/21 cases	[115]
		Positions 9, 13	miR-142-3p		
	BL	Position 4	miR-142-3p seed	1/5 cell lines	[114]
Leukemia	CLL	Positions 9, 14, 18, 20	miR-142-5p	1/452 cases	[119]
		Position 18	miR-142-3p		
		Position 7	miR-142-3p seed		
		Positions 41, 44	pre-miR-142		
		Positions 9, 14	miR-142-5p	1/90 cases	[120]
		Position 41	pre-miR-142		
		Position 14	miR-142-5p	1/210 cases	[121]
		Position 7	miR-142-3p seed		
	AML	Positions 2, 3, 4	miR-142-3p seed	1/416 cases	[122]
		Position 5	miR-142-3p seed	1/200 adult cases	[123]
		Positions 3, 6	miR-142-3p seed	2/200 adult cases	
		Position 3	miR-142-3p seed	1/149 cases	[114]

Abbreviations: Diffuse large B-cell lymphoma (DLBCL); Follicular lymphoma (FL); Burkitt lymphoma (BL); Chronic lymphocytic leukemia (CLL); and Acute myeloid leukemia (AML).

**Table 2 cells-13-00084-t002:** The aberrant function of mature miR-142-3p and/or -5p strands and their validated target(s) associated with carcinogenic roles in different hematological cancer types.

Cancer Type	Cancer Subtype	Mir-142 Strand	Mutation/Altered Expression in miR-142	Validated Direct Cancer Targets/Relevant Pathway	Identified Affected Function	Reference
Non-Hodgkin lymphoma	DLBCL	-3p	Loss-of-function mutation	*RAC1*/RAC1 pathway	Increased cell proliferation; decreased cell apoptosis	[116,124]
	FL	-3p	Loss-of-function mutation	*RAC1*/RAC1 pathway	Increased cell proliferation; decreased cell apoptosis	[116,118,124]
	BL	-5p	Upregulated expression in EBV+	*PTEN*/PI3K-AKT pathway	Decreased cell apoptosis	[132,133]
	MALT Lymphoma	-5p	Upregulated expression in H. *pylori* eradication resistant	*TP53INP1*	Decreased cell apoptosis	[135]
	NKTCL	-3p	Downregulated expression in EBV+	*IL1A*, *AC9*; *RICTOR*/IL-1; PI3K-AKT pathway	Increased cell growth, mobility, and proliferation; decreased cell apoptosis	[143,146,150,153,154]
	ATLL	-3p	Downregulated expression	*THBS4*	Increased tumor angiogenesis, cell migration, and vascular invasion	[156]
Leukemia	CML	-3p; -5p	Downregulated expression	*CIAPIN1*/tyrosine kinase-Ras pathway	Decreased cell apoptosis	[157,158]
		-5p	Downregulated in imatinib nonresponsive patients	*ABL2*, *MCL1*, *cKIT*, *SRI*/GM-CSF, IL-3, PI3K-AKT, JAK-STAT, Src family kinase, MAPK pathways	Increased cell invasion, metastasis, proliferation, cytoskeleton rearrangement, and imatinib and multi-drug resistance; decreased cell apoptosis	[159,160,161]
	ALL	-3p	Upregulated expression in T-ALL	*AC9*, *GRα*/cAMP-PKA pathway	Increased cell proliferation; GC resistance	[150,162]
		-3p	Upregulated expression in pre-B-ALL	*GRα*	Increased GC resistance	[162,163,164]
		-3p	Downregulated expression in t(4;11)(q21;q23) B-ALL	*MLL-AF4*	Increased cell proliferation; decreased cell apoptosis	[165]
	AML	-3p	Loss-of-function mutation	*ASHL1*	Loss of lymphoid potential, maintenance of myeloid potential, expansion of myeloblasts and immature hematopoietic cells (in IDH2^R140Q^/IDH2^R172K^+)	[27]
	APL	-3p	Downregulated expression	*CCNT2*, *TAB2*	Decreased monocytic and granulocytic differentiation	[36]

Abbreviations: Diffuse large B-cell lymphoma (DLBCL); Follicular lymphoma (FL); Burkitt lymphoma (BL); Mucosa-associated lymphoid tissue (MALT) lymphoma; Nasal natural killer/T-cell lymphoma (NKTCL); Adult T-cell leukemia/lymphoma (ATLL); Chronic myeloid leukemia (CML); Acute lymphocytic leukemia (ALL); Acute myeloid leukemia (AML); and Acute promyelocytic leukemia (APL).

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
