# Peer review of "miR-142: A Master Regulator in Hematological Malignancies and Therapeutic Opportunities"

_cells, 2023, doi:10.3390/cells13010084_

Round 1

Reviewer 1 Report

Comments and Suggestions for Authors

The review is very complete and of major interest for the field. It is very enjoyable to read and clearly highlight the complexity of the miRnas field. Showing first, how one specific microRnas has multiple organs and genes targets and how is function can be different depending of the cells and the circumstances. Then the authors give an extensive focuse on hématology normal hématopoiesis and malignancies. Finishing by browsing the different new therapeutic opportunities that the miRnas field potentially will open in the near future.

For me, the text can be accepted in its present form.

I have only a small suggestion concerning the division into subtitles, which can be more rigorous:

You wrote

2. The Role of miR-142 in Physiology and Hematology 

2.1. Biogenesis, Regulation, and Functions of miR-142 

I suggest to use only

2. Biogenesis, Regulation, and Functions of miR-142 

then you wrote

3. Physiological Functions of miR-142

I suggest

3. Physiological functions of miR-142 and its dysregulation apart from hematology 

3.1 circadian circle

3.2 bone 

3.3...

then

4. Hematopoietic Functions of miR-142

4.1 hématopoiesis

4.2 B cells

4.3...

5. Exosome-Mediated Transport of miR-142

6. The Role of miR-142 in Hematological Malignancies

.....

Author Response

Point-by-point response to reviewers’ comments

First of all, we thank the reviewers for their constructive comments. We extensively revised the text and addressed all the points raised by the reviewers.

Reviewer #1

1) I have only a small suggestion concerning the division into subtitles, which can be more rigorous:

Response: Now, we have worked on the subtitles and reorganized all of them.

Reviewer 2 Report

Comments and Suggestions for Authors

Overall, the work titled "miR-142: Master Regulator in Hematologic Cancers and Therapeutic Avenues" provides a comprehensive overview of the role of miR-142 in hematologic cancers and potential therapeutic strategies targeting this microRNA. The review covers a wide range of topics, including the biology of miR-142, its dysregulation in various hematologic cancers, and different therapeutic approaches. However, there are several aspects that could be improved or clarified in the paper:

  1. The paper is quite extensive, and the presented information could be better organized to enhance readability. Some content is blurry, while others are too technical without the context of "miRNA in hematologic cancers." Including fairly extensive sections unrelated directly to the topic, such as the description of the physiological role of miRNA in various organs, poses a challenge in the context of a scientific review. Expanding the topic into areas not directly related to the main aim of the paper, i.e., miR-142 in hematologic cancers, may lead to reader distraction. This can make it difficult to grasp key points and conclusions related to the main theme. Scientific work should focus on providing accurate and relevant information related to the research topic. Expanding content into unrelated areas can lead to information overload and content fragmentation.

  2. The role of miR-142 has been presented rather one-dimensionally in the paper, through the classical approach of "mutation in the gene - reduced molecule expression - impact on gene expression." The aspect of complex epigenetic regulation, including functional loops in which this molecule may be involved, has been overlooked. Introducing miR-142 into gene regulation in the context of epigenetic loops can provide a deeper understanding of its mechanisms in these cancers. It has been repeatedly demonstrated that miR-142 has the ability to directly influence gene expression by binding to the mRNA sequences of target genes. However, miR-142 can also impact epigenetic regulation, which is a significant element in controlling carcinogenesis processes. Firstly, miR-142 can affect the activity of epigenetic enzymes such as DNA methyltransferases or histone deacetylases. Through its interactions with these enzymes, miR-142 can influence DNA methylation and histone activity, which in turn affects chromatin accessibility and gene expression. This is particularly relevant in the case of tumor suppressor genes, whose activity may be inhibited by hypermethylation of their promoters. Additionally, miR-142 can participate in the formation of epigenetic loops, where it contributes to the maintenance of its own expression. In other words, miR-142 can regulate itself by influencing epigenetic chromatin modifications in the vicinity of its locus. This creates a form of regulatory loop that sustains the stability of miR-142 expression. Discussing these aspects, especially the impact of miR-142 on epigenetic regulation and its involvement in epigenetic loops, could complement the understanding of its role as a "master regulator" in hematologic cancers. This may provide additional mechanisms and points of therapeutic intervention worth investigating and utilizing in the future. I suggest that future studies and research consider these aspects and provide a more detailed discussion of their significance in the context of hematologic cancer biology.

  3. Although the review provides a comprehensive overview of miR-142, it could benefit from a more critical analysis of existing literature. Are there controversies or knowledge gaps regarding the role of miR-142 in hematologic cancers that require attention?

  4. The paper mentions mechanistic aspects of miR-142, but it would benefit from delving deeper into specific pathways and molecular mechanisms through which miR-142 exerts its actions. This would help readers better understand the underlying biology.

  5. The section on therapeutic approaches is informative, but could be more critical. Are there any limitations or challenges associated with the discussed therapeutic strategies that need highlighting? The paper briefly mentions FDA-approved therapies related to miR-142. Expanding on the clinical relevance of these therapies and their outcomes in patients would strengthen the review. It would be beneficial to prepare a comparative table presenting the latest trends in clinical studies using antagomiRs and mimics.

  6. The conclusions should include a concise summary of the main findings from the review and potentially suggest directions for further research or unresolved questions in the field.

  7. Carefully proofread the paper for language and grammatical errors to improve overall readability.

Comments on the Quality of English Language

The overall language quality in the manuscript is good, but there are some instances of awkward phrasing and word choice that could be improved for clarity and readability.

Author Response

Point-by-point response to reviewers’ comments

First of all, we thank the reviewers for their constructive comments. We extensively revised the text and addressed all the points raised by the reviewers.

Re

1) The paper is quite extensive, and the presented information could be better organized to enhance readability. Some content is blurry, while others are too technical without the context of "miRNA in hematologic cancers." Including fairly extensive sections unrelated directly to the topic, such as the description of the physiological role of miRNA in various organs, poses a challenge in the context of a scientific review. Expanding the topic into areas not directly related to the main aim of the paper, i.e., miR-142 in hematologic cancers, may lead to reader distraction. This can make it difficult to grasp key points and conclusions related to the main theme. Scientific work should focus on providing accurate and relevant information related to the research topic. Expanding content into unrelated areas can lead to information overload and content fragmentation.

Response: We have revised sections that describe the physiological role of miRNA in various organs. These sections have been removed from the text to avoid distracting the reader.

2) The role of miR-142 has been presented rather one-dimensionally in the paper, through the classical approach of "mutation in the gene - reduced molecule expression - impact on gene expression." The aspect of complex epigenetic regulation, including functional loops in which this molecule may be involved, has been overlooked. Introducing miR-142 into gene regulation in the context of epigenetic loops can provide a deeper understanding of its mechanisms in these cancers. It has been repeatedly demonstrated that miR-142 has the ability to directly influence gene expression by binding to the mRNA sequences of target genes. However, miR-142 can also impact epigenetic regulation, which is a significant element in controlling carcinogenesis processes. Firstly, miR-142 can affect the activity of epigenetic enzymes such as DNA methyltransferases or histone deacetylases. Through its interactions with these enzymes, miR-142 can influence DNA methylation and histone activity, which in turn affects chromatin accessibility and gene expression. This is particularly relevant in the case of tumor suppressor genes, whose activity may be inhibited by hypermethylation of their promoters. Additionally, miR-142 can participate in the formation of epigenetic loops, where it contributes to the maintenance of its own expression. In other words, miR-142 can regulate itself by influencing epigenetic chromatin modifications in the vicinity of its locus. This creates a form of regulatory loop that sustains the stability of miR-142 expression. Discussing these aspects, especially the impact of miR-142 on epigenetic regulation and its involvement in epigenetic loops, could complement the understanding of its role as a "master regulator" in hematologic cancers. This may provide additional mechanisms and points of therapeutic intervention worth investigating and utilizing in the future. I suggest that future studies and research consider these aspects and provide a more detailed discussion of their significance in the context of hematologic cancer biology

Response: While there is limited research available in the literature regarding the epigenetic regulation of miR-142, we have included a brief section on this topic in “Concluding Remarks and Outlook” section.

3) Although the review provides a comprehensive overview of miR-142, it could benefit from a more critical analysis of existing literature. Are there controversies or knowledge gaps regarding the role of miR-142 in hematologic cancers that require attention?

Response: We have expanded the " Concluding Remarks and Outlook" section and incorporated additional discussions pertaining to recent discoveries concerning miR-142 and identified gaps in our understanding of miR-142

4) The paper mentions mechanistic aspects of miR-142, but it would benefit from delving deeper into specific pathways and molecular mechanisms through which miR-142 exerts its actions. This would help readers better understand the underlying biology.

Response: We have expanded the information in the text regarding miR-142's target genes and associated signaling pathways, including PTEN, IL1A, PI3K/AKT, and SRI.

5) The section on therapeutic approaches is informative, but could be more critical. Are there any limitations or challenges associated with the discussed therapeutic strategies that need highlighting? The paper briefly mentions FDA-approved therapies related to miR-142. Expanding on the clinical relevance of these therapies and their outcomes in patients would strengthen the review. It would be beneficial to prepare a comparative table presenting the latest trends in clinical studies using antagomiRs and mimics.

Response: We have included the following paragraph in the text to address the limitations and challenges associated with the discussed miRNA-based therapeutic strategies. “Immune-related side effects also present an obstacle for the implementation of miRNA-based therapeutics [218]. This is evidenced in the clinical trial of MRX34 (a miR-34 mimic in a liposome) for the treatment of refractory tumors, which was halted in phase I after severe immune-related side effects led to the death of four patients [219]. Serious adverse events occurred later on in the treatment cycle after the conclusion of daily MRX34 infusions; these events included sepsis, hypoxia, cytokine release syndrome, and hepatic failure, which are symptoms characteristic of immune toxicity [219]. The underlying mechanism of these side effects has yet to be isolated. However, the absence of such responses in other oligonucleotide drugs utilizing the same liposome delivery method and the similarity of the side effects to those seen in patients undergoing immune checkpoint inhibitor therapy point to immune-mediated mechanisms driven by MRX34 [219]. Despite these challenges, the potential of miRNA mimics has led to their continued development.”

6) The conclusions should include a concise summary of the main findings from the review and potentially suggest directions for further research or unresolved questions in the field.

Response: We have updated conclusions section based on the reviewer’s comment.

7) Carefully proofread the paper for language and grammatical errors to improve overall readability.

Response: The Authors and Scientific Editor Department of Scientific Publications have reviewed and edited text.

Round 2

Reviewer 2 Report

Comments and Suggestions for Authors

The manuscript titled "miR-142: A Master Regulator in Hematological Malignancies and Therapeutic Opportunities" provides a comprehensive review of the role of miR-142 in hematological cancers. The authors, Wilson Huang, Doru Paul, George A Calin, and Recep Bayraktar, have made significant revisions in response to the constructive comments from the reviewers. Sections describing the physiological role of miRNA in various organs have been removed to maintain a focused approach on miR-142 in hematologic cancers. While acknowledging the limited research on the epigenetic regulation of miR-142, the authors have included a brief section on this topic in the "Concluding Remarks and Outlook" section.The suggested aspects, such as the impact on epigenetic regulation and involvement in epigenetic loops, are now discussed to complement the understanding of miR-142 as a "master regulator." The "Concluding Remarks and Outlook" section has been expanded to include a more critical analysis of existing literature. The authors discuss controversies, recent discoveries, and identified gaps in understanding miR-142 in hematologic cancers. The manuscript now delves deeper into specific pathways and molecular mechanisms through which miR-142 exerts its actions. This includes information on target genes and associated signaling pathways, such as PTEN, IL1A, PI3K/AKT, and SRI. The authors have added a paragraph addressing limitations and challenges associated with miRNA-based therapeutic strategies. A discussion on the immune-related side effects of miRNA-based therapeutics, illustrated by the case of MRX34, has been included to provide a critical perspective on these approaches. The conclusions section has been updated to include a concise summary of the main findings from the review.The authors suggest directions for further research and highlight unresolved questions in the field, enhancing the completeness of the manuscript.

Authors have diligently addressed the reviewers' comments, resulting in a more focused, well-organized, and informative manuscript on miR-142 in hematological malignancies. The revisions contribute to the overall quality and depth of the review, providing valuable insights.